



# Monitoring Arctic thin ice: A comparison between Cryosat-2 SAR altimetry data and MODIS thermal-infrared imagery

Felix L. Müller[1], Stephan Paul[2,1], Stefan Hendricks[2], and Denise Dettmering[1]

[1]Technical University of Munich, Germany; TUM School of Engineering and Design, Department of Aerospace & Geodesy, Deutsches Geodätisches Forschungsinstitut (DGFI-TUM)
[2]Alfred Wegener Institute, Helmholtz Centre for Polar and Marine Research, Bremerhaven, Germany

**Correspondence:** Felix L. Müller (felix-lucian.mueller@tum.de)

**Abstract.** Areas of thin sea ice in the polar regions are not only experiencing the highest rate of sea-ice production but are, therefore, also important hot spots for ocean ventilation as well as heat and moisture exchange between the ocean and the atmosphere. Through co-location of (1) Moderate Resolution Imaging Spectroradiometer (MODIS) derived thin-ice thickness estimates with (2) an unsupervised waveform classification (UWC) approach and (3) Sentinel-1 A/B SAR reference data, thin-
ice based waveform shapes are identified, referenced, and discussed with regard to a manifold of waveform shape parameters. Here, a strong linear dependency is found that shows the possibility to either develop simple correction terms for altimeter ranges over thin ice or to directly adjust current retracker algorithms specifically to very thin sea ice. This highlights the potential of CryoSat-2-based SAR altimetry to reliably discriminate between thick sea ice, open-water leads, as well as thin-ice occurrences within recently refrozen leads or mere areas of thin sea ice. Furthermore, a comparison to the ESA Climate
Change Initiative's (CCI) surface-type classification reveals that the newly found thin-ice related waveforms are divided up between almost equally between 'unknown' (46.3 %) and lead-type (53.4 %) classifications. Overall, the UWC results in far fewer 'unknown' classifications (1.4 % to 38.7 %). Thus, UWC provides more usable information for sea-ice freeboard and thickness retrieval while UWC at the same time reduces range biases from thin-ice waveforms processed as regular sea ice in the CCI classification.

## 1 Introduction

Areas of thin sea-ice cover in the polar regions play an important role for sea-ice production as well as the ventilation and heat exchange of the ocean with the atmosphere – especially during polar night time (e.g., Meier et al., 2014; Morales Maqueda et al., 2004; Maykut, 1978; Thorndike et al., 1975). Several studies investigated the distribution and occurrence frequencies of thin sea ice in sea-ice leads and sea-ice polynyas within the Arctic (e.g., Rothrock et al., 1999; Preußer et al., 2016; Preußer
et al., 2019; Tian-Kunze et al., 2014; Huntemann et al., 2014; Willmes et al., 2010; Willmes and Heinemann, 2015, 2016; Reiser et al., 2020) using different sensors and retrieval methodologies. However, the resulting data sets are generally bound to an upper sea-ice thickness limit in their retrieval capabilities as well as methodological limitations (e.g. cloud cover presence when using thermal-infrared data (e.g., Yu and Rothrock, 1996; Frey et al., 2008) or the spatial resolution using passive-microwave data (e.g., Tamura et al., 2008)).



In contrast, satellite altimetry is capable of retrieving sea-ice freeboard and sea-ice thickness for the Arctic on a basin-scale using CryoSat-2 and its predecessor sensors with several data products already available (e.g., Landy et al., 2019; Paul et al., 2018; Guerreiro et al., 2017; Kurtz et al., 2014). However, studies suggest a higher uncertainty towards thinner sea-ice (Ricker et al., 2017). This is the result of several factors. The radar backscatter characteristics between young and thin sea ice is ambiguous with open-water area, since both surfaces have specular reflection properties that are also prone to lead to off-

nadir reflection biasing the radar range (e.g., Aldenhoff et al., 2019; Passaro et al., 2018a; Rinne and Similä, 2016). However, classifying radar waveform echoes by their respective backscattering surface type (i.e., sea ice, open ocean, or leads) is essential for the upstream process of retrieving sea-ice freeboard and sea-ice thickness using satellite radar altimetry (e.g., Laxon, 1994; Laxon et al., 2003; Peacock and Laxon, 2004). Between converted surface-height estimates from sea-ice-type and lead-type waveform echoes one can subsequently calculate the sea-ice freeboard (i.e., the height differences between the sea-ice surface

and the ocean surface). Assuming hydro-static equilibrium and utilizing additional auxiliary information on sea-ice type and respective sea-ice density as well as information on the sea-ice snow cover one can calculate the sea-ice thickness (e.g., Paul et al., 2018; Alexandrov et al., 2010).

But even when a radar waveform is correctly classified as sea ice, the small freeboard values of young ice are often lower than the precision of even the later synthetic aperture radar (SAR) altimeter sensors. In addition, freeboard estimates over sea

ice for Ku-Band radar altimeters must be adjusted for a lower wave propagation speed in the snow layer (Mallett et al., 2020). In the absence of a direct observation for each radar altimeter waveform, snow depth and density information may originate from a climatology, modelled snow depth with re-analysis as input or data fusion of different satellite sensors. However, all approaches provide an average snow depth for a certain period and region, which will overestimate the snow layer on young sea ice in most cases and create a freeboard bias since the range correction depends on snow depth.

Hence, there is currently a need for additional satellite products or physical waveform models to better understand retrievals based on radar altimeter waveforms over thin-ice areas to complement improvements for thicker and rougher sea ice surfaces (e.g., Landy et al., 2019).

While information on the presence of thin-ice areas is important for our understanding on sea-ice mass balance changes, there is currently only a single operational thin-ice data product available due to the above-mentioned short-comings and limitations.

This product is based on the ESA's Soil Moisture and Ocean Salinity (SMOS) mission (Tian-Kunze et al., 2014) based on the ice-thickness dependency of surface emissivity at L-Band, however, at a lower spatial resolution of $12.5 \, \text{km} \times 12.5 \, \text{km}$. This method is limited for thicker sea ice and thus data fusion of SMOS and CryoSat-2 using Optimal Interpolation (Ricker et al., 2017) is routinely used to improve sea-ice thickness retrieval over the full range of the sea ice thickness distribution. Data products utilizing thermal-infrared data to estimate thin-ice thickness and corresponding areas are also available, but not in a

similar operational fashion (e.g. Preußer et al., 2016; Preußer et al., 2019).

In this study, the authors utilize Delay-Doppler radar altimeter echoes from ESA's Earth Explorer mission CryoSat-2 in combination with the capabilities of NASA's Moderate Resolution Imaging Spectroradiometer (MODIS) to monitor thin ice with a high spatial-temporal resolution. This allows for a better understanding of the received CryoSat-2 waveform returns over thin sea-ice areas and, subsequently, an improved surface-type classification. For this task, the authors inter-compare





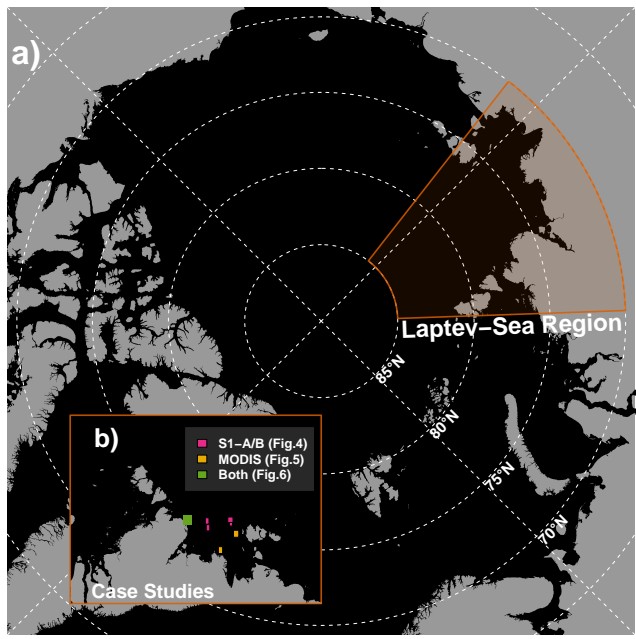

**Figure 1.** Overview of the study area within the Arctic Laptev-Sea region (a; orange outline). Small rectangles within the map inlet (b) mark the locations of case studies used in this study (see Figures 4 to 6).

CryoSat-2 waveforms labelled as thin ice through an unsupervised classification approach (extended from Müller et al. (2017)) with thin-ice thickness estimates from MODIS (Paul et al., 2015) within a maximum of 30 minutes between both acquisitions. All investigations are performed for the Arctic Laptev-Sea region featuring frequent occurrences of flaw and coastal polynyas (Fig. 1; e.g. Preußer et al., 2019; Willmes et al., 2010) and cover the winter months January through March between 2011 and 2020.

This study is structured into the following subsections: Section 2 describes all the data sets used followed by Section 3 providing details on the employed unsupervised clustering for CryoSat-2 as well as the MODIS thin-ice thickness retrieval. Finally, Section 4 summarizes and discusses the study's results and implications on CryoSat-2 surface-type classification and sea-ice thickness estimation and concludes with an outlook (Section 5).

## 2 Dataset

The following sub-section highlights all data sets used for this study. All analyses are carried out for the Arctic Laptev Sea region between 92 °E to 142 °E and 70 °N to 85 °N (see Fig. 1) for the winter months January through March between 2011 and 2020.



## 2.1 CryoSat-2 Level-1B Baseline-D data

CryoSat-2 was launched in April 2010 aiming at the monitoring of the Earth's cryosphere, in particular, the thinning of sea
ice and the rise of sea levels in the polar seas. CryoSat-2 was placed on a non-Sun-synchronous orbit with a long repeat
orbit of about 369 days. Moreover, CryoSat-2 is carrying a Ku-Band radar altimeter and is observing the cryosphere in three
different acquisition modes up to a latitude of $88\,^\circ$ (Scagliola, 2013). The acquisition modes vary with application area and
surface type with respect to a geographical mode mask (more information see: https://earth.esa.int/eogateway/instruments/
siral/description). The polar seas, mainly characterized by a seasonally changing sea ice cover, are predominantly sampled by
the Synthetic Aperture Radar (SAR) mode showing an along-track footprint size of about 300 meters (Wingham et al., 2006).
Briefly summarized, the SAR mode combines frequency-shifted radar returns (i.e. Doppler effect) in the direction of flight
from different look angles with respect to a predefined position on the surface. The result of combining different view angles
is called multi-look waveform and samples the surface with 20Hz resolution (about 250 m along-track resolution).

The present investigation is based solely on SAR-mode acquired multi-looked waveforms, covering major parts of the
Arctic Ocean. In particular, Cryosat-2 L1B Ice Baseline D data are introduced to the processing chain. This data comprises
in addition to the waveform data also information waveform scaling as well as orbit positions. More information regarding
Baseline D can be found in the ESA Cryosat-2 Product Handbook (https://earth.esa.int/eogateway/documents/20142/37627/
CryoSat-Baseline-D-Product-Handbook.pdf/c76df710-2a5c-c8b8-00c1-13c8db0e9f51, last-access: October, 2021) and Mel-
oni et al. (2020).

## 2.2 MODIS data

As basis for the comparison of thin-ice thickness to Cryosat2 waveform returns, Moderate Resolution Imaging Spectroradiome-
ter (MODIS) Level 1B calibrated radiances are obtained from both MODIS sensors on board the polar-orbiting NASA satellites
Terra and Aqua (MOD/MYD02; MODIS Characterization Support Team (MCST), 2017a, b; retrieved from the LAADS DAAC
at https://ladsweb.modaps.eosdis.nasa.gov/, last access: 06.07.2021) with a spatial resolution of $1\,\text{km} \times 1\,\text{km}$ at nadir and swath
dimensions of $1354\,\text{km}$ (across track) $\times\,2030\,\text{km}$ (along track).

In a first step, brightness temperatures were calculated from the calibrated radiances comprising MODIS channels 31 and
32 following Toller et al. (2009). Subsequently, the sea-ice-surface temperature (IST) was computed following Riggs and Hall
(2015). All MODIS processing is based on MODIS Collection 6.1 data.

In order to compute corresponding thin-ice thickness data from the IST data, additional data on the prevailing atmospheric
conditions are necessary. Here, all necessary atmospheric fields are provided from the European Centre for Medium-Range
Weather Forecast (ECMWF) ReAnalysis V5 (ERA5) data acquired from the Copernicus Climate Data Store (CDS; Hersbach
et al., 2020). These fields comprise the $2\,\text{m}$ air temperature, the $10\,\text{m}$ wind-speed components, the mean sea-level pressure,
and the $2\,\text{m}$ dew-point temperature.



### 2.3   Sentinel-1A/B SAR images

For an improved understanding of the CryoSat-2/MODIS comparison the authors utilize images from the side-looking ESA Copernicus C-Band SAR missions Sentinel-1A and B (S1-A/B). All acquired S1-A/B scenes were used for additional visual comparison as they are unaffected by cloud cover. Furthermore, S1-A/B features a higher spatial resolution with regard to MODIS and, therefore, provide further information about different sea-ice-surface types by making use of the backscattering properties of different surfaces in the ice covered ocean. For instance, leads and polynyas appear very dark due to a very flat

and less rough surface. In this case the incoming radar signal is scattered away from the receiver. However, caution must be taken by the interpretation of the backscattering pixel values in the presence of small scale features like frost flowers (i.e small ice crystals with a size of few centimeters in diameter) that develop under cold and calm conditions, e.g. on nilas ice, they can significantly increase the scattering resulting in brighter pixel values (Hollands and Dierking, 2016). More information regarding sea-ice-surface-type interpretation can be found in Dierking W. (2013).

Sentinel-1A and Sentinel-1B were launched in 2014 and 2016, respectively, and orbit the Earth 180° apart on a six-day repeat cycle in a two-satellite configuration. In the present study, the Sentinel-1 comparison dataset consists of Level-1 dual-polarized SAR extra-wide-swath mode data at medium resolution. The images are ground-range detected and show a pixel resolution of 40 m and a swath width of 400 km. The images are processed using the SNAP - ESA Sentinel Application Platform v8.0, (http://step.esa.int) following the processing steps described in Müller et al. (2017) and Passaro et al. (2018b), but with an

additional speckle filtering. The Level-1 data is gathered from the Alaska Satellite Facility Data Active Archive Center (ASF DAAC).

### 3   Methods

The following section focuses on the key steps of data processing that enable a comparison of thin-ice observations from SAR altimetry and MODIS thermal-infrared imagery. In order to reduce the influence of rapid changing environmental conditions

such as sea ice drift, rapid melt as well as freezing periods, the maximum time gap between altimetry, thermal-infrared imagery and SAR imaging is set to ±30 minutes around the overflight times of CryoSat-2. This is a compromise to minimize observation situation changes, while at the same time keeping a sufficiently large MODIS as well as SAR image database. This is necessary due to the frequent presence of cloud cover in the MODIS data.

The MODIS comparison database used includes 161 scenes, which corresponds to an altimetry dataset of about 21300

Cryosat-2 observations (MODIS scenes and their time difference from the Cryosat-2 observations are given out in a list upon request). The number of usable observation pairs of both missions is dependent on cloud-free conditions, the processed thin-ice area and the availability of SAR mode data from Cryosat-2.



## 3.1 Cryosat-2 thin-ice classification

Müller et al. (2017) developed an unsupervised classification of altimeter waveforms that was primarily focused on the detec-
tion of open water targets, such as leads and polynyas, without the use of training data or pre-known information about the
surface conditions. The classification can be applied to any kind of altimeter waveforms and missions. Müller et al. (2017)
applied it to the LRM missions Envisat and SARAL. Later, the classification approach was adopted to SAR altimeter wave-
forms (Dettmering et al., 2018) and applied to Cryosat-2 and Sentinel-3A/B in the framework of the European Space Agency's
Baltic+ Sea Level (ESA Baltic SEAL) project (Passaro et al., 2021). However, none of these studies has classified thin-ice,
since altimeter waveforms generated by this surface type are quite similar to open water returns.

Briefly summarized, the unsupervised waveform classification (UWC) is based on an automatic clustering of a broad spectra
of different waveforms by K-medoids (e.g. Celebi (2014)) defining a reference model with a certain number of clusters. After
clustering next classification steps consist of an assignment of the clustered waveforms as well as remaining waveforms to
certain surface types (e.g. ocean, lead/polynya or sea-ice conditions). In the present investigation, the cluster number is set to
25. Following Dettmering et al. (2018), by using this number an overall agreement of about 97 % can be achieved. Moreover,
an internal miss-classification rate of 1.13 % can be expected after performing a 10-fold cross validation (Passaro et al., 2020).

Main input to the classification approach are parameters derived from the waveform shapes and information on the backscat-
ter power. Altogether, they span the so-called feature space, which is defined as followed (see Dettmering et al. (2018)):

– **Maximum power (MP)** - physical backscatter coefficient $\sigma_0$ at the waveform maximum

– **Waveform width (Wwidth)** - number of waveform range bins with a power greater than 1% of the waveform maximum

– **Leading edge slope (LES)** - number of waveform range bins between the position of the waveform maximum and the
bin of the leading edge, which is the first exceeding 12.5% of the maximum power

– **Trailing edge slope (TES)** - similar to LES, but for the trailing edge of the waveform

– **Waveform decay (Wdecay)** - estimation of the decay by fitting an exponential function to the trailing edge

– **Waveform fit Median-Absolute Deviation (WfitMAD)** - derived MAD value of the residuals from the fit of the expo-
nential function

Moreover, in the quantitative comparison (Sec. 4.2) two additional features are included, which are not used for the UWC:

– **Leading edge width (LEW)** - width in range bins between 95% and the first bin at the leading edge exceeding 5% of
the waveform maximum power using a ten-time over-sampled and smoothed waveform (Hendricks et al., 2021)

– **Leading edge peakiness (LEP)** - computation of the pulse-peakiness (Peacock and Laxon, 2004), but three range bins
left from the bin position of the waveform maximum (Ricker et al., 2014)



Figure 2 shows the assignment of 25 clusters to 5 different surface types (compared to 4 in the original UWC approach): undefined, sea ice, thin-ice, lead, and ocean. Clusters characterized by a very strong maximum power (MP), a small waveform width (Wwidth) and strong decay obtained by the fitting of an exponential function to the trailing edge of the waveform are labeled as lead clusters (i.e. 5, 10, 14). Clusters that were formerly assigned to sea ice are now divided into two groups (sea ice and thin ice) based on their different reflective properties. Following Ulander et al. (1995) and Onstott and Shuchman (2004) thin ice covers nila and young ice ranging from 0 m to 0.30 m and is often covered by a flat layer of slush, which are ice crystals saturated with salt brine. Members of thin-ice clusters show characteristics in between of lead clusters and clear sea-ice groups. They are characterized by a wider waveform shape, a weaker waveform decay and flatter trailing edge slope than lead clusters. Moreover, they also have a slightly weaker maximum power than radar returns reflected from open, calm water (Zygmuntowska et al., 2013). These properties correspond to clusters 4 and 19.

"Undefined" waveform clusters represent waveforms which cannot assigned unambiguously to certain surface conditions, for example if they are acquired in the direct vicinity of the coast or islands. They also show a bigger standard deviation compared to other clusters.

## 3.2 MODIS thin-ice thickness retrieval

In order to compute the thin-ice thickness (TIT) from MODIS ice-surface temperatures (IST) a simple surface-energy-balance model is employed, which utilizes the inversely proportional relation between IST and the thickness of thin sea ice (Yu and Rothrock, 1996; Drucker et al., 2003). In the model, the net positive flux towards the atmosphere (i.e., positive corresponding to the direction from the warm ocean to the cold atmosphere) is equalized from the conductive heat flux through the ice. From this conductive heat flux, TIT is derived following Equation 1, where $TIT$ is the thin-ice thickness, $\kappa_i$ is the thermal conductivity of sea ice ($2.03\,\mathrm{W(m\,K)}^{-1}$), $IST$ and $T_{fp}$ are the ice-surface temperature and the ice/ocean-interface temperature (assumed to be at the freezing point temperature of the ocean), respectively, and $Q_{atm}$ is the total heat flux to the atmosphere.

$$TIT = \kappa_i \times \frac{IST - T_{fp}}{Q_{atm}} \qquad (1)$$

A detailed description of the TIT retrieval procedure as well as all necessary equations and related assumptions are thoroughly described in Paul et al. (2015). For ice thicknesses between 0.0 and 0.2 m, Adams et al. (2012) state an average uncertainty of $\pm 4.7\,\mathrm{cm}$, substantially improving for larger thickness ranges. Figure 3 exemplifies the underlying IST with its corresponding TIT. With regard to these uncertainty limitations in combination with the desire to maximize CryoSat-2/MODIS overlaps, all thin-ice thickness analyses are limited to a maximum sea-ice thickness of 25 cm.

## 4 Results and discussion

In this study, a combination of 161 MODIS swaths covering the Laptev-Sea area showing TIT up to 25 cm and about 21300 classified CryoSat-2 observations are used for a quantitative analysis. The spatio-temporal very high resolution altimetry obser-



**Figure 2.** Averages and standard error of 6 waveform features used per cluster: Maximum Power (MP); Waveform Width (Wwidth); Leading-Edge Slope (LES); Trailing-Edge Slope (TES); Waveform decay (Wdecay) and median absolute deviation of fitted waveform (WfitMAD). Colors indicate 5 different surface types: "undefined" (brown), "sea ice" (yellow), "thin ice" (orange), "lead" (cyan) and "ocean" (blue).

vations are related to the respective MODIS pixels using a nearest-neighbor approach. Moreover, Sentinel-1A/B SAR images serve as an additional source for visual comparisons. The first part of this section shows visual comparisons between CryoSat-2 and MODIS as well as Sentinel-1. The second part will then focus on a quantitative analysis of the results.





## 4.1 Visual comparison

This section provides an initial visual inter-comparison between CryoSat-2 classified open-water/leads, thin-ice, and sea-ice observations on the one hand and either MODIS (Table A1) or Sentinel-1A/B SAR imagery (Table A2) on the other hand. Moreover, comparison with operational SMOS products are performed.

Figure 4 shows four zoomed-in snippets of two HH-polarized SAR images from February 2018 with their respective classification result from CryoSat-2 superimposed on them. Both SAR images feature at least one lead, which are recognized by the classification algorithm and highlighted by the cyan dots. Orange labels indicate thin-ice observations in the vicinity of leads and correspond to higher SAR backscatter values. Within the image, the areas appearing in light gray to almost white with their lead-like pattern represent former open water areas that have recently frozen over. These areas of new ice can range in thickness from 10 cm to 30 cm (Onstott and Shuchman, 2004) and are often covered by a layer of slush (Ulander et al., 1995). The CryoSat-2 classification results agree reasonably well with the SAR images, however, not all thin ice surfaces and leads are always correctly detected (e.g. Fig. 4 c,d). Potential reasons for this are two-fold: From the SAR point of view, a complete interpretation of the SAR pixel values is not possible and, therefore, no truth is available. From the altimetry point of view, off-nadir effects may overlay clear leads or thin-ice radar echoes, which hence prevents a clear identification, in particular if the lead or thin-ice surface is very small. Since the observations are very close together in time, differences due to ice movement or over-freezing can be excluded.

A visual comparison between MODIS TIT estimates and thin-ice assigned CryoSat-2 radar echoes are provided by Figure 5. Both sets of images are recorded in January 2014 and 2017, respectively. White areas to the North indicate either a lack of data due to present cloud cover or an ice-thickness estimate above 25 cm. Towards the South, thin-ice areas are bounded by either the coast line and/or the extensive presence of landfast sea ice (Preußer et al., 2019; Selyuzhenok et al., 2015; Dmitrenko et al., 2005). Northwards leaving the respective coast-line or fast-ice edges, MODIS TIT features a rather steady increase in ice thickness. The respective CryoSat-2 classification generally agrees with respect to the MODIS TIT, showing primarily lead and thin-ice classifications. Qualitatively, the respective classifications appear within expectations, in particular

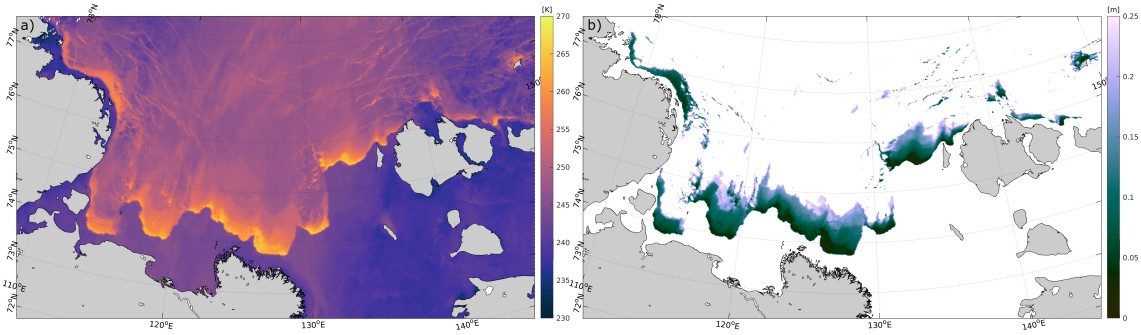

**Figure 3.** Example of MODIS ice-surface temperature (IST in K; a) and its associated thin-ice thickness (TIT; b) between 0 m and 0.25 m; acquired on 05-03-2011 at 18:10 UTC.

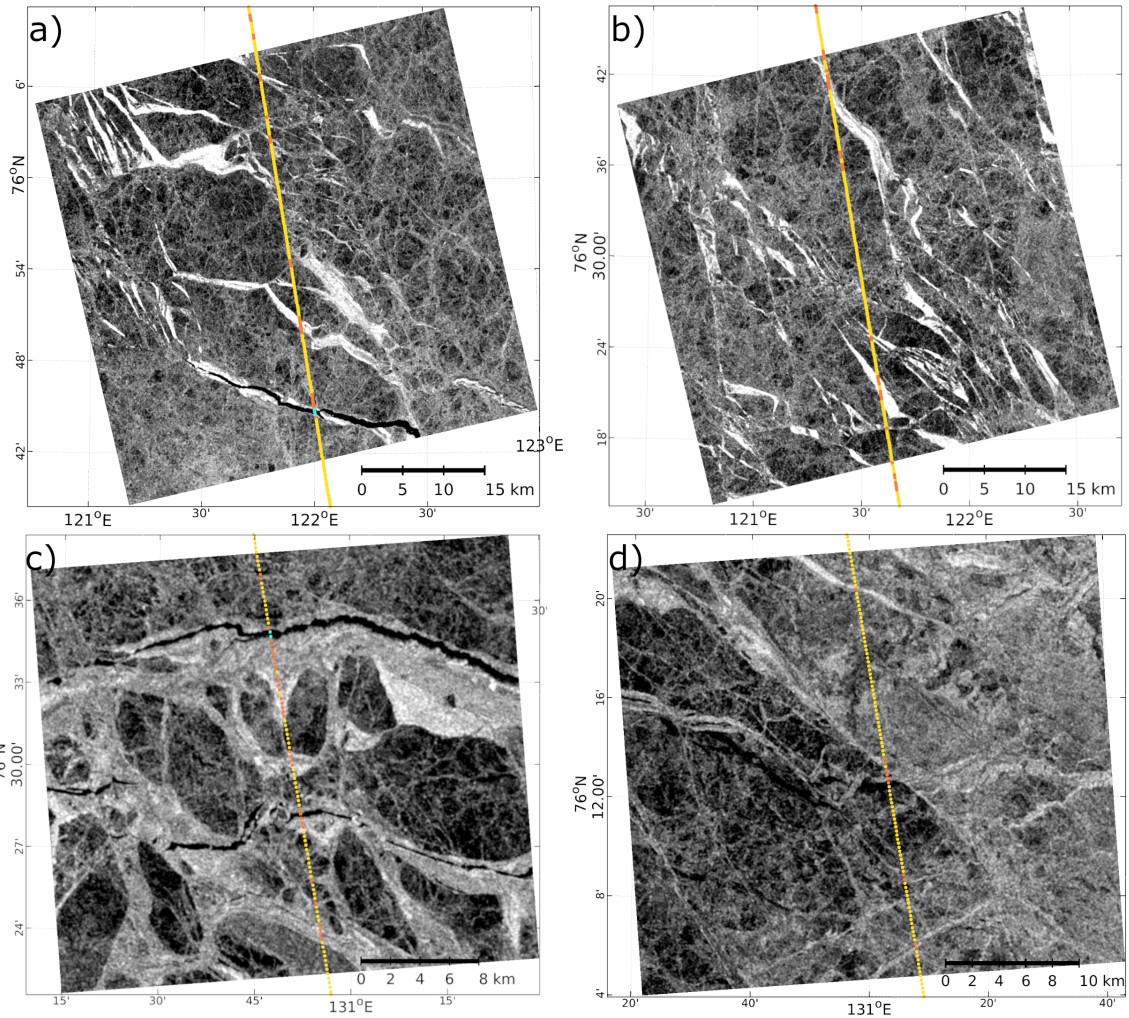

**Figure 4.** Examples of classified CryoSat-2 observations versus HH-polarized Sentinel-1B (a,b) and Sentinel-1A (c,d) snapshots from February 2018 with acquisition time gaps of about 2 minutes and 1 minute, respectively. Orange markers show thin-ice classifications. Leads are labeled in cyan, sea ice in yellow. Figures are north-oriented.

with lead classifications close to the coast-line or fast-ice associated ice edge in the South where the sea ice is thinnest and thin-ice classifications further North, where MODIS features thicker thin-ice estimates (see Figure 5, c,d). However, a direct

distinction between thicker sea-ice and thin-ice areas is not possible due to the coarse pixel resolution of MODIS in comparison to CryoSat-2. Nevertheless, a general connection between thin-ice-labeled radar returns in the direct vicinity of thin sea-ice can be recognized. This relationship is investigated more deeply in a quantitative analyses using the full MODIS database.

Figure 6 displays a CryoSat-2, MODIS, and Sentinel-1A comparison acquired at 01 March 2018 featuring an acquisition time gap within 17 minutes from each other (more details can be found in the Appendix). This gives the very rare opportunity to





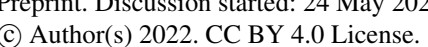

**Figure 5.** Examples of classified CryoSat-2 observations versus MODIS TIT acquired in January, 2014 (a,b) and January, 2017 (c,d), respectively. The left side shows an overview, whereas a detailed view, indicated by the purple rectangle, is given on the right side. Orange markers show thin-ice classifications. Leads are labeled in cyan. Figures are north-oriented.

analyse observations of all three sensors within a 30 minutes time frame. It shows the different behaviour and dependencies of optical and microwave sensors with respect to different sea-ice surfaces and their physical properties (e.g. surface roughness).

     The scene shows very thin-ice close to the landfast-ice edge near Taimyr in the Northeastern part of the Laptev-Sea region (Dmitrenko et al., 2005; Selyuzhenok et al., 2015). The polynya observed by MODIS appears very bright in the SAR image and is supposedly caused by a rough sea-ice surface due to the presence of frost flowers (Hollands and Dierking, 2016; Dierking





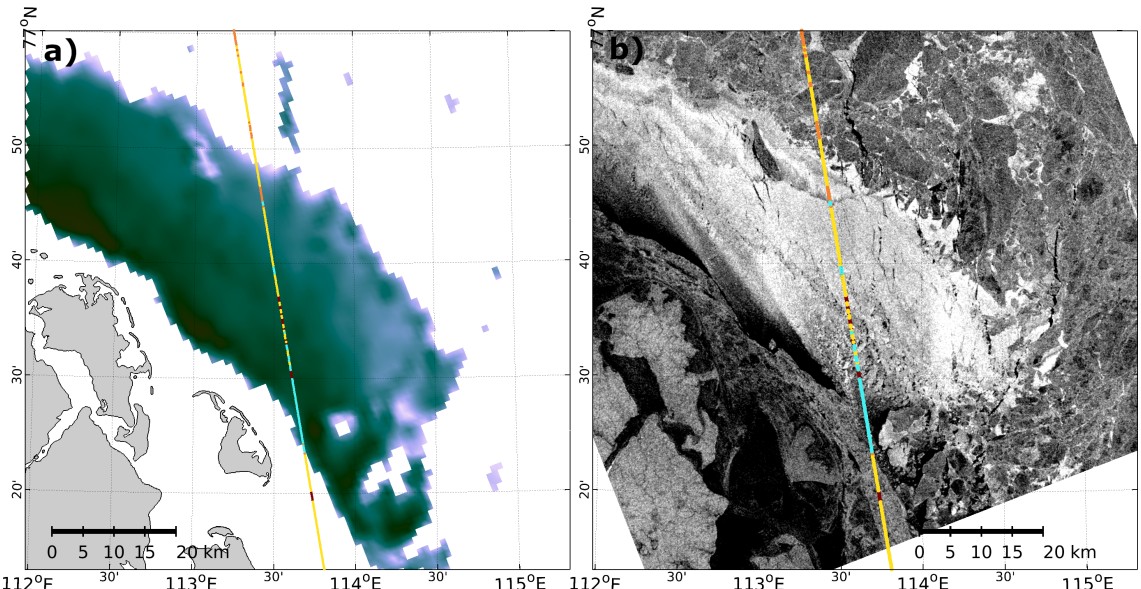

**Figure 6.** Examples of classified CryoSat-2 observations versus MODIS TIT (a) and Sentinel-1A SAR (b) for the same location within a time gap of 7 minutes (MODIS) and 24 minutes (Sentinel-1A) w.r.t Cryosat-2, respectively. Orange markers show thin-ice classifications. Leads are labeled in cyan. Figures are north-oriented.

W., 2013). In contrast to the impact this has on the side-looking SAR, the increased surface scattering results in a reduced backscattering power leading to a noisy radar reflection for the nadir-looking CryoSat-2. This results in a false assignment of sea ice within the classification scheme. However, the low backscatter region to the Southeast (dark in the SAR image) featuring very thin ice are labeled correctly again as lead. While the CryoSat-2 unsupervised classification performs in general really well, it was primarily developed to identify open water within sea-ice conditions. Hence, a clear distinction between thin

and thick sea ice is not always possible due to a high variability of surface roughness of thin-ice. This comparison showcases how substantially the sea-ice surface can vary within small spatial scales but at the same time the enormous potential and synergies for sea-ice investigations that lies within these multi-sensor collocations.

      For further visual comparison, SMOS-derived sea-ice thicknesses (Tian-Kunze et al., 2014) are acquired through the sea-ice portal data portal (https://www.meereisportal.de) and blended with CryoSat-2 classified waveforms.

SMOS-derived sea ice thickness depicts daily composites with a reference time of 12:00:00Z, therefore, CryoSat-2 passes of $\pm 12\,\mathrm{hours}$ around this reference time are selected. In order to account for the much coarser spatial resolution of $12.5\,\mathrm{km} \times 12.5\,\mathrm{km}$ for SMOS, CryoSat-2 classifications are aggregated into bins with a length of $12.5\,\mathrm{km}$. This is done once for the fraction of CryoSat-2 thin-ice classifications as well as for the fraction of CryoSat-2 classifications of either thin-ice and lead. Due to the low spatial resolution, this type of comparison is of course not ideal. However, SMOS-derived sea-ice thickness is to date the

only operational product delivering reasonable thin-ice thickness estimates.



Two SMOS comparisons from March 2014 and March 2017 during the occurrence of large polynyas in the Laptev Sea show good agreement between thin-ice areas, but also leads (Fig. 7). For example, a larger proportion of lead observations are found in areas featuring very thin ice. CryoSat-2 thin-ice classifications are predominantly found in areas up to a SMOS-derived ice thickness of 25 cm. However, thin-ice observations are also frequently present in areas with thicker ice according to the SMOS data. This is likely a result of the comparable low resolution of the SMOS data, being unable to fully resolve fine lead structures as well as the long aggregation period of the SMOS data over a full day. This long aggregation period tends to reduce the impact of fast-growing very thin sea ice. Vice versa, CryoSat-2 sea-ice classifications are also present in SMOS thin-ice areas. The reason for this stems from the fact that the thin-ice classification of Cryosat-2 is mainly focused on recently frozen or re-frozen leads (i.e., single-peak like waveforms).

Overall, the rapidly changing environmental conditions in the Arctic region and the high sea-ice growth rates, especially for very thin ice during winter, make a detailed comparison and interpretation difficult. Especially, since the SMOS sea-ice thickness data represent only an averaged/aggregated snapshot over a full day, this likely results in a smoothing effect in particular for the thinnest sea-ice fractions. Therefore, a quantitative comparison with SMOS-derived sea-ice thickness is not considered further.

## 4.2 Quantitative analysis

For the quantitative analysis, a total of about 21,300 CryoSat-2 observations are compared with MODIS TIT observations in the Laptev Sea for the winter months January through March between 2011 and 2020. This corresponds to about 4 % of the theoretically available matched MODIS vs. CryoSat-2 along-track observations (total circa: 540,000). The small percentage of matches used is explained by 1) the frequent presence of cloud cover in the Arctic, as well as 2) the availability of large flaw-lead polynya openings with corresponding thin-ice surfaces, which occur near the landfast sea-ice edge.

In total, 14 % of all 21300 classifications were assigned to thin-ice surfaces and 1 % to leads. The largest proportion is attributed to thicker sea-ice (85 %). With respect to valid overlaps (i.e. MODIS pixels with TIT information), about 51 % of Cryosat-2 classifications are assigned to open water and thin-ice, while 45 % are assigned to other sea-ice types. Remaining waveforms are marked as "undefined", e.g., due to the influence of land or instrument errors.

Besides a comparison of the CryoSat-2 classification results and the thin-ice thickness from MODIS, the relationship between waveform-derived backscatter power as well as other waveform-shape properties and different thin-ice thickness categories is of peculiar interest. Therefore, the thin-ice thickness data from MODIS is grouped into intervals of ±2 cm from 4 cm to 25 cm. These groups are then compared to spatiotemporally corresponding CryoSat-2 radar waveforms, in particular, with the six waveform-derived features used in the classification and two additional waveform parameters (i.e. LEW and LEP) listed in Section 3.1.

Figure 8 shows the per-bin averages of the 8 waveform features with their respective standard deviation indicated as gray bars in the background. With the exception of LES, there is a linear (or close to linear) dependency present w.r.t. an increasing thin-ice thickness.
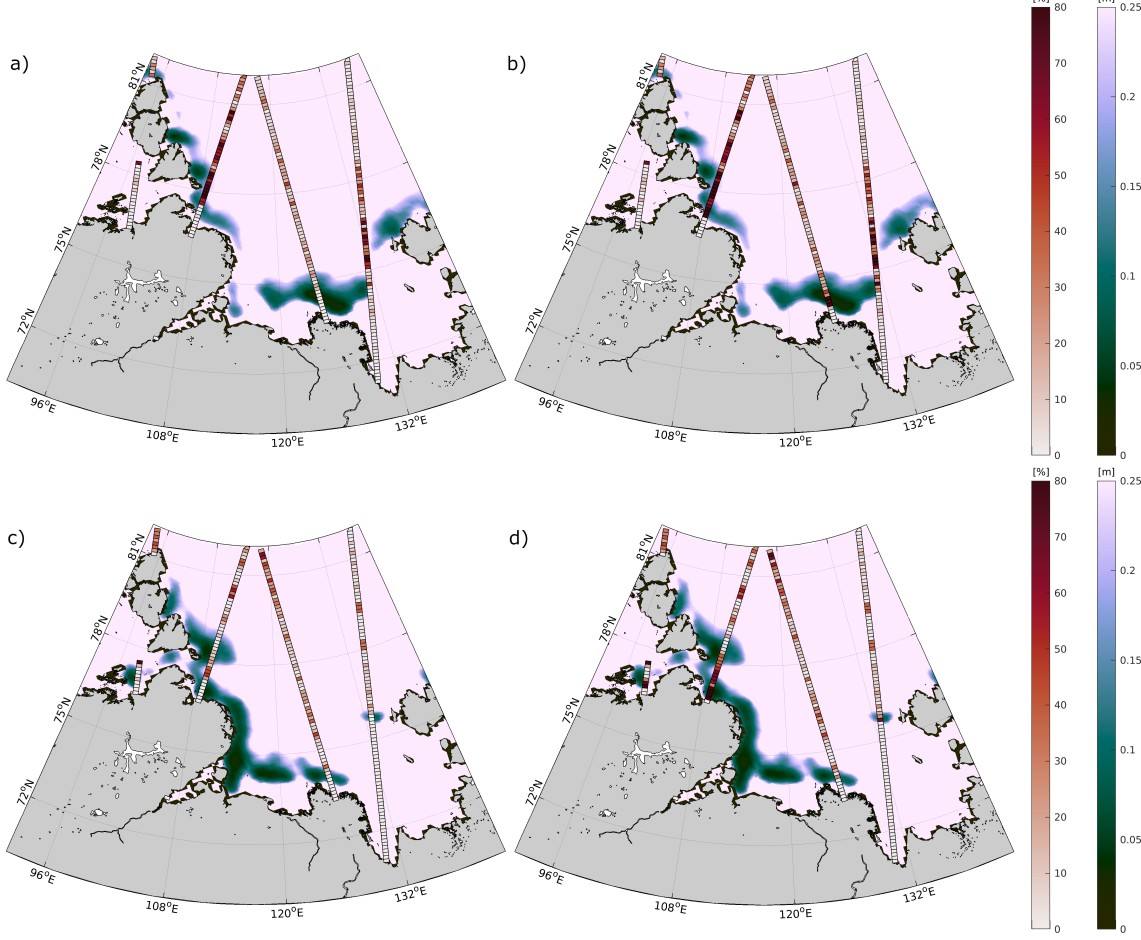

**Figure 7.** Percentage of thin-ice only (a,c) and thin-ice + lead (b,d) classifications per bin (length 12.5 km) with SMOS sea ice thickness limited to 25 cm from 2014-03-23 (a,b) and 2017-03-30 (c,d) in the background. Figures are north-oriented.

This is especially evident in waveform features MP, Wwidth, and TES that are intended to characterize single-peak wave-
forms (e.g. radar reflections by a lead). Despite a very large standard deviation in the thinner ice-class bins, an increase with
rising ice thickness is a physically explainable behaviour. In general, a rougher surface corresponds to an increase of surface
scattering resulting in a diminishing return back to the sensor and at the same time an increase of received off-nadir scatter-
ing by the sensor. This leads to a broadening of the received waveform as well as a drop in maximum waveform power (e.g.
Drinkwater, 1991; LAXON, 1994) and, therefore, to the observed negative correlation with MP and the positive correlations
with TES and Wwidth.

However, in the case of LES, which is the number of bins between the first waveform bin reaching 12.5 % and the bin of
the waveform's maximum power (Dettmering et al. (2018)), this linear relationship cannot be spotted. This might be due to an



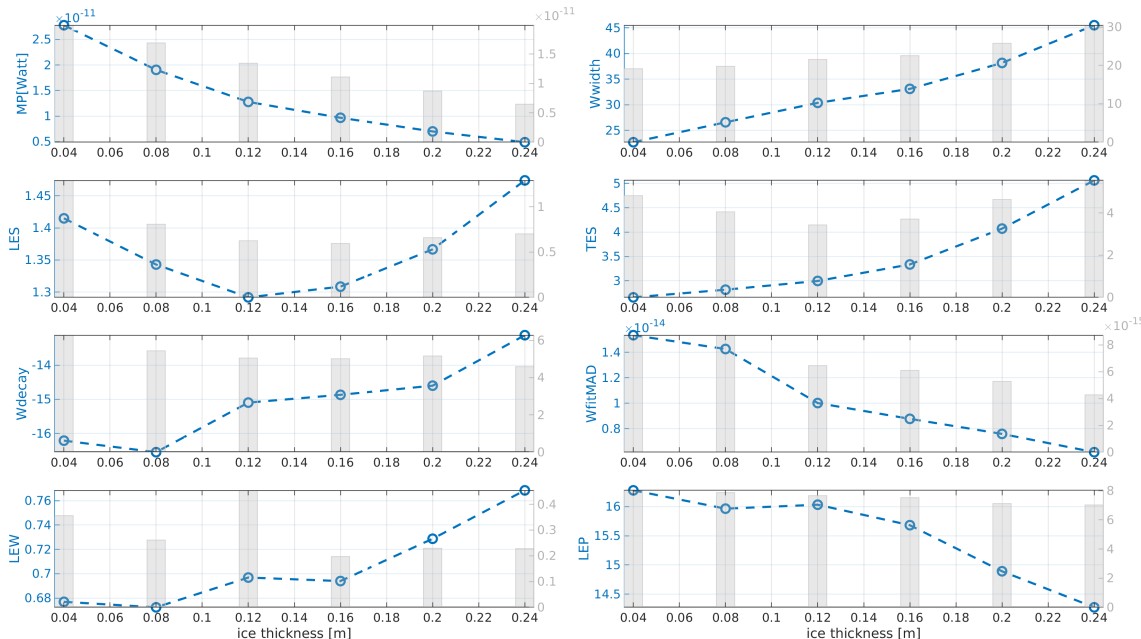

**Figure 8.** Waveform-derived shape and power features from unsupervised classification plus two external features Leading-Edge-Width (LEW; Hendricks et al. (2021)), Leading-Edge-Peakiness (LEP; Ricker et al. (2014)) with respect to different thin-ice thickness categories. Shown are the averaged feature values (blue dotted line) and the corresponding standard derivation (gray bar) per thin-ice thickness groups of $\pm 2\,\mathrm{cm}$ from 2 to 25cm. Abbreviations of waveform features used in classification are explained in Fig.2

.

enhanced error budget in the first two thin-ice thickness groups or higher uncertainties in the LES processing related to very steep leading edges. Only the latter part, starting with the $\pm 12\,\mathrm{cm}$ bin features the expected behaviour.

The remaining four waveform features, i.e., Wdecay, WfitMAD, LEW, and LWP, feature an apparent correlation, however, it is less clear than the ones we observed with MP, Wwidth, and TES. Similar to TES, physically one would expect a slower/smaller decay with an increase in surface roughness and therefore sea-ice thickness as well as an increase in the width of the leading edge (LEW, Landy et al., 2019) from thinner to thicker ice classes. Contrariwise, the leading-edge peakiness (LEP) is expected to decrease with an increasing leading-edge width going along with an increase surface roughness. The

mean-absolute deviation from the fitted exponential fit to the waveforms is also decreasing for broader waveforms that result from addition received backscatter from off-nadir areas due to an increased surface roughness.

    Nevertheless, this comparison provides information on how different thin-ice scenarios affect the altimeter waveform. The shown dependencies between TIT and derived waveform parameters can be used, e.g., to adapt and optimize altimeter waveform classifiers to thin-ice conditions and subsequently improve the sea-ice freeboard estimation in areas of frequently present

thin ice.



## 4.3 Classification Comparison

We compare the surface type classification with the CryoSat-2 surface classification described in Paul et al. (2018) to assess how thin-ice-class waveforms are represented in other CryoSat-2 based products. The classification by Paul et al. (2018) was developed for the CryoSat-2 contribution to the sea-ice-thickness data record of the ESA Climate Change Initative (citation),
hereafter named CCI classification. The algorithm is also used in the AWI CryoSat-2 sea-ice product v2.4 (Hendricks et al., 2021) which provides the necessary temporal coverage for this study. We use Level-2 intermediate (l2i) data, which provides the surface-type flag for full resolution orbit data for the months of October through April from November 2010 to April 2021 for the Arctic Laptev sea regions (Fig. 1). The flags in the l2i files are based on monthly thresholds for the backscatter coefficient sigma0, the leading-edge width, and pulse peakiness as well as supported by sea-ice-concentration data as sea-ice
mask. The surface types used in Paul et al. (2018) are comparable to this study except for the missing thin-ice class and the fact that the ocean-waveform classification solely depends on the sea-ice mask and existing land/ocean flags.

The surface-type classification is based on approximately 26.5 million waveforms. The CCI classication lists 50.2 % of these in the sea ice category, 10.9 % in the lead category, 38.7 % in the unknown, and less then 0.1 % in the ocean category (Fig. 9a). Compared to the CCI classification, the unsupervised waveform classification algorithm (UWC) of this study has
more sea ice (79.7 %, +29.2 %) and ocean (3.9 %, +3.89 %) waveform, as well as less lead-type (1.9 %, -9 %) and unknown waveforms (1.4 %, -37.3 %) in addition to the 13.0 % thin-ice waveforms (Fig. 9b). The classification matrix (Fig. 9c) shows that the UWC thin-ice waveforms are mostly distributed in CCI unknown (46.3 %) and CCI lead (53.4 %) surface types with a negligible contribution of CCI sea-ice waveforms (0.3 %). UWC lead classifications are almost exclusively (96.8 %) in the CCI lead class, showing a good agreement in the identification of open-water leads. UWC sea-ice classification also has a good
agreement with CCI sea-ice classifications (61.9 %) but also include waveforms the CCI algorithm labels as unknown (35.6 %).

Between the two surface type classifications algorithms, the UWC results in far fewer unknown waveforms than the CCI algorithm and, thus, provides more usable information for sea-ice freeboard and thickness retrieval. The distinction of leads and thin ice by the UWC algorithm, which are partly seen as leads in the CCI algorithm, reduces the number of sea-surface height observations, but increases their reliability. It should be noted that we define a lead here in the sense of satellite altimetry
as an open-water lead, which provides a true sea-surface-height observation without any bias introduced by thin-ice freeboard. The sea-ice thickness bias introduced by using thin-ice freeboard as sea-surface height on the surrounding sea ice is in the order of the thin-ice thickness itself and with a maximum of 25 cm a significant fraction of typical first-year ice thickness. The fact that the CCI algorithm labels a significant portion of the UWC thin-ice waveforms as unknown, demonstrates that these were rightfully exluded from either lead or sea-ice class. The latter sea-ice class is treated as an ice surface, for which range
corrections for the full climatological snow cover are applied, which would also result in range biases. The distinction between open-water lead and thin-ice classes, therefore, introduces the possibility to improve both, the sea-surface height and radar freeboard retrieval.





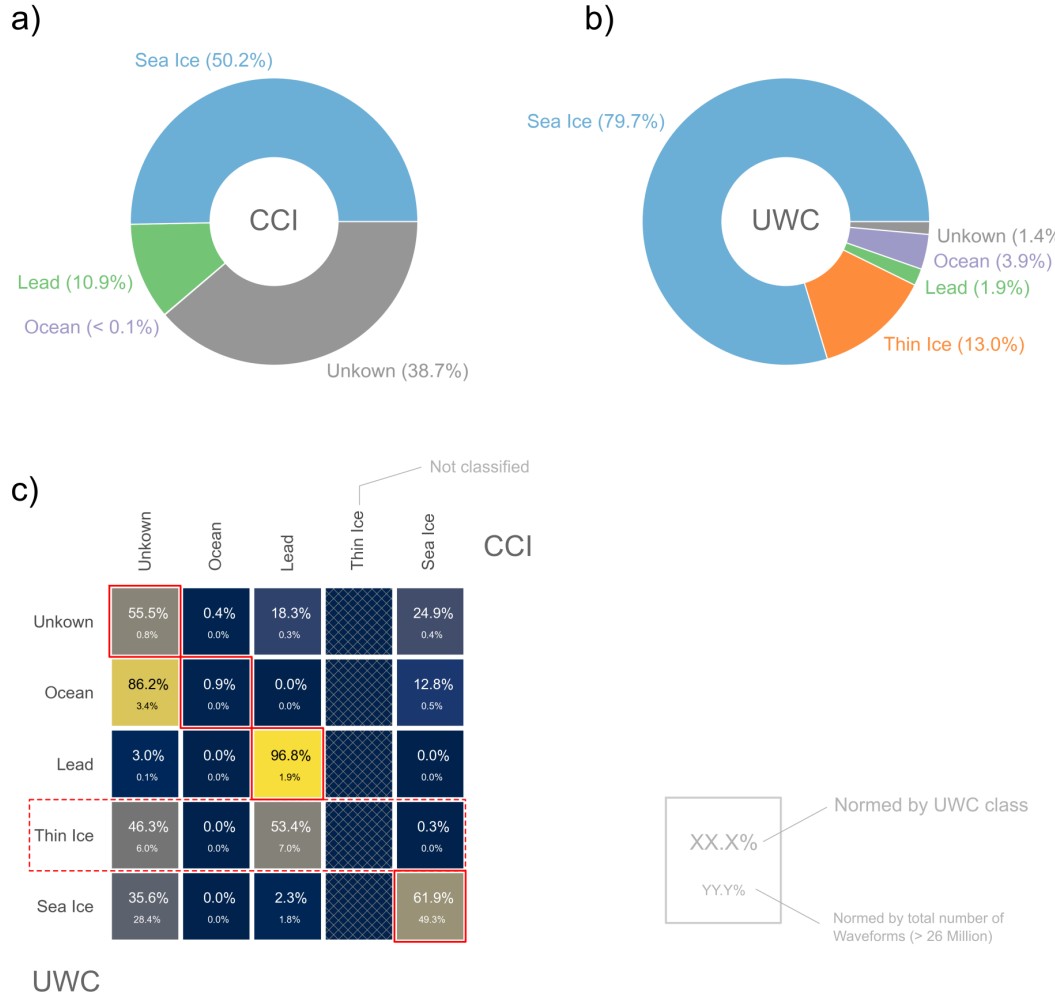

**Figure 9.** Surface-type-classification statistics of CryoSat-2 data for the Arctic Laptev Sea region (Fig. 1) for a) the ESA Climate Change Initiative (CCI) algorithm described in Paul et al. (2018), b) the unsupervised waveform classification (UWC) of this study and c) the classification matrix. The color coding of the classification matrix and upper percent values describes the distribution of UWC classes within CCI classes for each class combination. The lower percent values describes the fraction of the total number of waveforms in the comparison (> 26 million) in class combinations. The CCI surface type classification does not contain a thin ice class.

Finally, it must be noted that the official definition of the World Meteorological Organisation (WMO) for the term lead includes thin ice with a thickness of up to 30 cm (wmo, 2014). To obtain lead fractions according the WMO definition, or to
compare lead fractions with other remote sensing data, the thin-ice class may be added to the open-water lead class.



## 5 Summary and outlook

In the context of an increasing number of open-water areas and sea-ice thinning, which have a major impact on the energy exchange between the atmosphere, the upper-ocean layer and on sea-ice dynamics in the Arctic Ocean (e.g. Persson and Vihma (2017)), this study investigated the thin-ice detection capabilities of CryoSat-2 through comparison of altimetry-derived
thin-ice surface detections and thin-ice thickness information from MODIS thermal-infrared imagery. In addition, the study benefits from spatially and temporally consistent comparisons of three different remote-sensing techniques for continuous monitoring of the Arctic Ocean.

An unsupervised waveform classification approach (Dettmering et al., 2018; Müller et al., 2017), mainly developed to identify open-water targets such as leads and polynyas within the otherwise ice-covered ocean, is adopted to distinguish thin-
ice from water and thicker ice. Here, waveform clusters have been assigned to be of thin-ice type due to their resemblance to lead-type CryoSat-2 waveform echoes, but with in general less distinctness. These labeled CryoSat-2 observations are compared to thin-ice estimates up to 25 cm of thickness derived from MODIS thermal-imagery (Paul et al., 2015), to Sentinel-1 SAR images, to SMOS results (Tian-Kunze et al., 2014), as well as to an external classification approach (Paul et al., 2018; Hendricks et al., 2021).

A visual comparison shows an in general good agreement between the classified CryoSat-2 altimeter data and both, the Sentinel-1A/B SAR images as well as the MODIS-derived thin-ice areas despite different spatial resolutions as well as the varying delay in acquisition times between all sensors. However, especially the interpretation of the SAR images without ground-truth validation can be challenging in thin-ice areas due to a large variety of present surface conditions ranging from smooth or slushy areas (with low backscatter returns) to areas covered with frost flowers (with high backscatter returns).
The latter also affects the CryoSat-2 radar reflections, since higher surface roughness conditions result in a more sea-ice-like waveform shape with increased noise and weaker peak power. This potentially results in a sea-ice group assignment. Despite the substantial difference in spatial resolution between CryoSat-2 and MODIS, the majority of lead and thin-ice assigned CryoSat-2 observations are located in regions with very thin thin-ice. In case of SAR data, CryoSat-2 can help to bring thin-ice and lead-type information to a larger scale without the need to deal with substantial amounts of data and complicated processing
chains. With regards to MODIS, the CryoSat-2 thin-ice and lead-type can help to identify sub-pixel scale information from MODIS that results from spectral mixing of different surfaces within a single pixel.

In addition, a comparison was made between the Cryosat-2 waveform classification and SMOS-derived sea-ice thicknesses. For this purpose, the Cryosat-2 surface classifications were grouped into bins with a length of 12.5 km. Similar to MODIS, there is generally good agreement between thin-ice regions and thin-ice classifications. However, the coarse pixel resolution
and the daily averaging of SMOS observations do not allow deeper investigations. Moreover, it is shown that SMOS cannot resolve fine re-frozen lead structures. In contrast, sea-ice classifications also show up in SMOS-derived thin-ice regions. The reason for this might be the focus of the unsupervised classification on re-frozen leads. This fact brings the future task to check further waveform clusters regarding possible thin-ice detections. In this context the impact of wind and wave movements in

thin-ice area has to be investigated, which can likely cause a widening of the waveform while keeping a higher backscatter
power.

A quantitative comparison of CryoSat-2 waveform-derived shape and power features and MODIS-derived thin-ice thickness
shows a strong linear dependency with increasing thin-ice thicknesses, a finding that can be exploited for directly estimating
sea-ice thickness in a thickness regime where the freeboard-based approach lacks sensitivity. At a minimum it brings the
opportunity to use this information for adjusting retracker algorithms, such as for example the Threshold First Maximum
Retracker Algorithm (TFMRA, Helm et al. (2014)) to thin-ice conditions and to modify assumptions of snow cover in these
cases. It is also feasible that the obtained information can help to develop a correction term for altimeter ranges for thin-ice
waveforms based on the estimated thickness and well-known density of young sea ice for precise sea-level estimation in thin-
ice leads and open-water leads alike. Moreover, to enable further Arctic climate-relevant investigations using altimetry data the
presented classification of thin-ice can be extended to the entire Arctic to produce maps of altimetry-derived thin-ice coverage.

*Data availability.* CryoSat-2 Baseline-D L1B data (ESA, 2019) are freely available from the CryoSat-2 science server https://science-pds.
cryosat.esa.int (last access: October, 2021). ESA Copernicus Sentinel-1A/B Level-1 data is publicly available from ASF DAAC, (https:
//search.asf.alaska.edu/, last-access: October, 2021). MODIS Level 1B calibrated radiances obtained from the MODIS sensors on board
the polar-orbiting NASA satellites Terra and Aqua (MOD/MYD02; MODIS Characterization Support Team (MCST), 2017a, b) are freely
available from the Level-1 and Atmosphere Archive and Distribution System (LAADS) Distributed Active Archive Center (DAAC) at
https://ladsweb.modaps.eosdis.nasa.gov/, (last access: July, 2021). SMOS sea ice thickness from 2014-03-23 and 2017-03-30 were obtained
from https://www.meereisportal.de (grant: REKLIM-2013-04, last-access: March, 2022).

*Sample availability.* The MODIS image IDs of the comparisons are given out in a list upon request.

*Author contributions.* FM developed the classification approach and conducted major parts of the comparison. SP designed and performed
the thin-ice thickness estimation from MODIS thermal-infrared imagery. FM and SP drafted the original manuscript. SH contributed to the
quantitative comparison and writing of the manuscript and supported the study with discussions and comments. DD supported the study with
discussions of the applied methods and results and reviewed the manuscript.

*Competing interests.* The authors declare no conflict of interests.

*Acknowledgements.* The authors want to thank the LAADS DAAC and the ASF DAAC for the provision of the MOD/MYD02 and S1-
A/B data as well as the ECMWF and the CDS for the provision of the necessary ERA5 reanalysis data at no cost. Moreover, they want to





thank ESA for operating and managing the Earth Explorer Opportunity Mission Cryosat-2. The production of the here-used SMOS sea-ice-thickness data was funded by the ESA project SMOS & CryoSat-2 Sea Ice Data Product Processing and Dissemination Service, and data from 2014-03-23 and 2017-03-30 obtained from https://www.meereisportal.de (grant: REKLIM-2013-04).

This work was supported by the Technical University of Munich (TUM) in the framework of the Open Access Publishing Program.



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





Table A1: Shown MODIS scenes in Fig. 5 and Fig. 6, record date and time difference (Δ t) to Cryosat-2 observations in minutes (absolute value).

| MODIS ID | Record date | Δ t [min] |
|---|---|---|
| MOD03.A2014022.1250.061.2017308004553 | 2014-01-22 | 8.5 |
| MYD03.A2017031.1730.061.2018029032056 | 2017-01-31 | 9.3 |
| MYD03.A2018060.2300.061.2018061160404 | 2018-03-01 | 6.9 |

Table A2: Shown SAR images in Fig. 4 and Fig. 6, record date and time difference (Δ t) to Cryosat-2 observations in minutes (absolute value).

| SAR image ID | Record date | Δ t [min] |
|---|---|---|
| S1B_EW_GRDM_1SDH_20180211T231533_20180211T231633_009582_01143A_7062 | 2018-02-11 | 1 |
| S1A_EW_GRDM_1SDH_20180218T221829_20180218T221929_020667_02365D_58AE | 2018-02-18 | 2 |
| S1A_EW_GRDM_1SDH_20180301T231607_20180301T231707_020828_023B73_E32D | 2018-03-01 | 24 |