# Peer review of "Monitoring Arctic thin ice: A comparison between CryoSat-2 SAR altimetry data and MODIS thermal-infrared imagery"

_The Cryosphere, 2022_

## Referee Comment (RC1)

**Monitoring Arctic thin ice: A comparison between Cryosat-2 SAR altimetry data and MODIS thermal-infrared imagery**

Felix L. Müller, Stephan Paul, Stefan Hendricks, and Denise Dettmering

**4 July 2022**

**General comments**

The authors have conducted a study which is related to on-going development of sea ice thickness (SIT) retrieval algorithms for CryoSat-2 (CS2) radar altimeter. They modified an existing unsupervised CS2 waveform classification (UWC) method by Müller et al. (2017) to include thin ice class as an additional output (previous sea ice class in now divided into sea ice and thin ice classes). The new UWC was applied to the CS2 data over the Laptev Sea region (as shown in Figure 1) for the winter months Jan-Mar in 2011-2020. The C2 classification results were compared to MODIS thin ice thickness (TIT; max 25 cm) swath charts (in total 161), few Sentinel-1 (S-1) SAR images. SMOS SIT charts, and to ESA CCI CS2 surface classification product which does not have thin ice class. The comparisons are conducted by visual and quantitative analyses. The results suggest (as summarized in Abstract) that there is possibility to either develop simple correction terms for CS2 ranges over thin ice or to directly adjust current retracker algorithms specifically to very thin sea ice. Further, the new UWC can rather reliably discriminate between sea ice, open-water leads, and thin ice within recently refrozen leads or mere areas of thin sea ice. The new UWC surface type classification should be better for freeboard and SIT retrieval than the current ESA CCI CS2 product, e.g. the new UWC gives much less unknown surface type classifications. It may be possible, after further developments, to estimate SIT of thin ice based on the CS2 waveform data (freeboard methods lacks sensitivity to very small freeeeboards of thin ice).

I think the study set up with data acquisitions and data processing is sound, as are methods for the data analyses. However, I think there are some room for improvements, and in the following I have some general questions and comments, followed by miscellaneous specific ones.

First, the you should clearly define in the Abstract and Introduction what you mean by 'thin ice', what is the maximum assumed thickness for it? I think it is not good to use term 'young ice' in this context, as in the WMO sea ice nomenclature it is a sea ice class with 10-30 cm thickness.

All the used datasets should be presented under Dataset Section, this is not the case now for SMOS SIT data and the ESA CCI product, e.g. the ESA CCI product is now described in Section 4.3.

The discussion on thin ice remote sensing methods in Introduction is quite brief and should be expanded. This can focus on altimeter data based studies and methods, but it is good to discuss also other ones, e.g. discussion on high frequency radiometer data based thin ice detection is very brief, there have be many studies on recent years, see e.g.:

K. I. Ohshima, S. Nihashi, and K. Iwamoto, "Global view of sea-ice production in polynyas and its linkage to dense/bottom water formation," Geosci. Lett., vol. 3, no. 1, p. 13, 2016.

K. Nakata, K. I. Ohshima, and S. Nihashi, "Estimation of thin-ice thickness and discrimination of ice type from AMSR-E passive microwave data," IEEE Trans. Geosci. Remote Sens., vol. 57, no. 1, pp. 263–276, Aug. 2019.

K. Nakata, K. I. Ohshima, and S. Nihashi, "Mapping of active frazil for Antarctic coastal polynyas, with an estimation of sea-ice production," Geophys. Res. Lett., vol. 48, no. 6, Mar. 2021, Art. no. e2020GL091353.

       First study about frazil ice monitoring.

It is nice that the authors are also using S-1 SAR imagery as reference data, but only with visual analysis. Currently, there are many automatic S-1 SAR sea ice classification algorithms available, e.g. this one especially for lead detection:

Murashkin et al., Method for detection of leads from Sentinel-1 SAR images, Annals of Glaciology (2018), doi: 10.1017/aog.2018.6

This algorithm was developed at UB and AWI, so maybe the authors have access to it? At least lead detection by SAR should be briefly discussed. See also S-1 SAR sea ice classification studies, e.g.:

Park, J.-W., Korosov, A. A., Babiker, M., Won, J.-S., Hansen, M. W., and Kim, H.-C.: Classification of sea ice types in Sentinel-1 synthetic aperture radar images, The Cryosphere, 14, 2629–2645, https://doi.org/10.5194/tc-14-2629-2020, 2020.

Boulze, H.; Korosov, A.; Brajard, J. Classification of Sea Ice Types in Sentinel-1 SAR Data Using Convolutional Neural Networks. Remote Sens. 2020, 12, 2165. https://doi.org/10.3390/rs12132165

Previous altimeter lead/thin ice detection studies should also be discussed in more detail, their methods and results summarized, and later to compare the methods/results of this study to the previous ones, maybe it would be good to add Discussion Section. Some previous related studies:

Lee et al., Arctic lead detection using a waveform mixture algorithm from CryoSat-2 data, The Cryosphere, 12, 1665–1679, 2018, https://doi.org/10.5194/tc-12-1665-2018

N. Longépé et al., "Comparative Evaluation of Sea Ice Lead Detection Based on SAR Imagery and Altimeter Data," in IEEE Transactions on Geoscience and Remote Sensing, vol. 57, no. 6, pp. 4050-4061, June 2019, doi: 10.1109/TGRS.2018.2889519.

Wernecke, A. and Kaleschke, L.: Lead detection in Arctic sea ice from CryoSat-2: quality assessment, lead area fraction and width distribution, The Cryosphere, 9, 1955–1968, https://doi.org/10.5194/tc-9-1955-2015, 2015

I guess their relevance to the study depends how 'leads' are defined in them, are they only open water covered or also may have thin ice cover? In your CS2 UWC classification how you do define differences between classes 'lead' and 'thin ice'? Your 'lead' has only open water surface, or could it also have few cm of thin ice coverage?

You could shortly summarize in Introduction the comparison results from:

Dettmering, D., Wynne, A., Müller, F. L., Passaro, M., and Seitz, F.: Lead Detection in Polar Oceans—A Comparison of Different Classification Methods for Cryosat-2 SAR Data, Remote Sensing, 10, https://doi.org/10.3390/rs10081190, 2018.

It is understandable to you have a limited study region, and the Laptev Sea with recurring polynyas is good for it, but it would have been nice to see how your new UWC performs over some MIZ, like Barents Sea where large areas of thin ice may occur at the ice edge.

AWI conducts HEM flights over the western Arctic, would it be possible to include also HEM data to your study? It would give accurate data on the leads and thin ice, although temporal and spatial matching with CS2 tracks is likely a problem. You could discuss about this in Introduction, even in case the HEM data are not added.

Did you investigate also using Sentinel-2 imagery as lead/thin ice reference data? It has been used in some sea ice studies, e.g.

Muchow, M., Schmitt, A. U., and Kaleschke, L.: A lead-width distribution for Antarctic sea ice: a case study for the Weddell Sea with high-resolution Sentinel-2 images, The Cryosphere, 15, 4527–4537, https://doi.org/10.5194/tc-15-4527-2021, 2021.

Petty, et al. (2021). Assessment of ICESat-2 sea ice surface classification with Sentinel-2 imagery: Implications for freeboard and new estimates of lead and floe geometry. Earth and Space Science, 8, e2020EA001491. https://doi. org/10.1029/2020EA001491

Why did you not use the latest CS2 L1B baseline-E data? https://earth.esa.int/eogateway/news/new-ice-baseline-e-and-near-real-time-processors

I have following questions on the MODIS data in Section 2.2 and MODIS TIT calculation in Section 3.2:

How cloud masking is conducted?

Are both daytime and nighttime MODIS images used?

Why did you not use the MOD29 product which has cloud masked IST?

What ERA5 fields were used, hourly? If so then reference for this is:

H. Hersbach et al. (2018). ERA5 Hourly Data on Single Levels From 1979 to Present. Copernicus Climate Change Service (C3S) Climate Data Store (CDS). Accessed: X. [Online]. Available: https://doi.org/10.24381/cds.adbb2d47

Why did you not use ERA5 mean surface downward long-wave radiation flux? How it was now obtained?

Is snow on thin ice neglected in the TIT calculation?

MODIS TIT accuracy is given according to Adams et al. (2012), but is your TIT calculated exactly as in this paper? If not then their accuracy figure may not be valid for your TIT.

Your description of new UWC in Section 3.1 does not include any information on the weather conditions. Your study was conducted in winter conditions, but still is there any dependence on resulting clusters on the air temperature? Thin ice surface condition: slushy, dry, frost flowers etc., depend on the weather conditions. What was the air temp range in your study? Any above zero cases?

You should summarize in Section 3.1 rules how the 25 clusters are classified to different surface classes, e.g. with table or flowchart. You could also show statistics and pdf's of some features for different classes. In general, your description how clusters are classified is now very rough.

On backscattering mechanism discussion in lines 230-231 and 281-283: you should include discussion on the coherent vs. non-coherent scattering for radar altimeter. The coherent scattering is very high, increasing roughness decreases it quickly and non-coherent scattering due to roughness increases.

In Section 4.2 Quantitative analysis the CS2 vs. MODIS comparison results should be presented also in a table/matrix format. Give also the accuracies for CS2 thin ice and thick ice classes against the MODIS TIT charts, and e.g. commission error or False Alarm Rate: percentage of MODIS thin ice pixels falsely classified as thick ice out of all MODIS thin ice pixels.

It would be interesting to also see how CS2 thin ice classification works as a function of MODIS TIT, a similar investigation is now conducted for CS2 waveform features.

In Summary and outlook Section: "This fact brings the future task to check further waveform clusters regarding possible thin-ice detections."

What this exactly means, that you are not sure yet what surface classes all your 25 clusters really represent? If not then you should conduct further studies already in this paper.

**Spesific comments**

Abstract

"the ESA Climate Change Initiative's (CCI) surface-type classification"

You should mention that this is also based on the CS2 data, and that it does not have thin ice class.

**1. Introduction**

page 1, line 21: "However, the resulting data sets are generally bound to an upper sea-ice thickness limit in their retrieval capabilities as well as methodological limitations (e.g. cloud cover presence when using thermal-infrared data"

It would be good to mention here also the effect of snow cover.

l. 27: "However, studies suggest a higher uncertainty towards thinner sea-ice (Ricker et al., 2017)."

Please give the current 'accurate' SIT retrieval range of CS2.

In the CS2 SIT retrieval discussion you could also mention the effect of snow salinity, decreases penetration depth:

Nandan, V., et al. (2017). Effect of snow salinity on CryoSat-2 Arctic first-year sea ice freeboard measurements. Geophysical Research Letters, 44. https://doi.org/10.1002/2017GL074506

l. 48: "While information on the presence of thin-ice areas is important for our understanding on sea-ice mass balance changes, there is currently only a single operational thin-ice data product available due to the above-mentioned short-comings and limitations."

JAXA has AMSR2 based operational research product "Detection of thin sea ice", see https://suzaku.eorc.jaxa.jp/GCOM_W/data/data_w_product-3.html

l. 52: "This method is limited for thicker sea ice and thus data fusion of SMOS and CryoSat-2 using Optimal Interpolation"

Give the maximum SIT for an 'accurate' SMOS SIT estimation, in papers by Kaleschke and Tian-Kunze et al.

l. 56: "In this study, the authors utilize Delay-Doppler radar altimeter echoes from ESA's Earth Explorer mission CryoSat-2 in combination with the capabilities of NASA's Moderate Resolution Imaging Spectroradiometer (MODIS) to monitor thin ice with a high spatial-temporal resolution."

To my understanding MODIS TIT data is used for comparison, and not in combination with CS2 for thin ice monitoring.

l. 62: "flaw and coastal polynyas"

Describe shortly what these are; good general info for readers.

**2. Dataset**

Or Datasets and their processing?

l. 87: "ESA Cryosat-2 Product Handbook (https://earth.esa.int/eogateway/documents/20142/37627/ CryoSat-Baseline-D-Product-Handbook.pdf/c76df710-2a5c-c8b8-00c1-13c8db0e9f51,"

Make this as reference.

l. 97: "2009). Subsequently, the sea-ice-surface temperature (IST) was computed following Riggs and Hall (2015)."

Original reference for MODIS IST is:

Hall, D.; Key, J.; Casey, K.; Riggs, G.; Cavalieri, D. Sea ice surface temperature product from MODIS. IEEE Trans. Geosci. Remote Sens. 2004, 42, 1076–1087.

In your discussion of S-1 SAR backscatter signatures for leads a good reference is:

Murashkin et al., Method for detection of leads from Sentinel-1 SAR images, Annals of Glaciology (2018), doi: 10.1017/aog.2018.6

which discusses about dark and bright leads. This paper could also be as your basis for visual lead identification in the SAR imagery (if you keep doing so, and not applying some automatic method). Give the total number of S-1 SAR images used in your study.

l. 117: "The images are ground-range detected and show a pixel resolution of 40m and a swath width of 400km."

   The pixel size is 40 by 40 m, and spatial resolution is close 100 m depending on range.

l. 118: "The images are processed using the SNAP - ESA Sentinel Application Platform v8.0, (http://step.esa.int) following the processing steps described in Müller et al. (2017) and Passaro et al. (2018b), but with an additional speckle filtering."

   Please summarize these processing steps.

**3. Methods**

l. 129: "The MODIS comparison database used includes 161 scenes, which corresponds to an altimetry dataset of about 21300 Cryosat-2 observations (MODIS scenes and their time difference from the Cryosat-2 observations are given out in a list upon request)."

   What is the spatial resolution of these CS-2 observations? Just the area for one CS2 waveform? And there can be several waveforms within one 1 km MODIS IST pixel?

l. 137: spell out LRM acronym.

l. 139: "However, none of these studies has classified thin-ice, since altimeter waveforms generated by this surface type are quite similar to open water returns."

   Can you at this point summarize the difference between the open water and thin ice backscatter/waveforms based on earlier studies?

l. 142: "After clustering next classification steps consist of an assignment of the clustered waveforms as well as remaining waveforms to certain surface types (e.g. ocean, lead/polynya or sea-ice conditions)."

   How this is conducted? I would assume some automatic method has been developed, please describe it.

l. 144: "In the present investigation, the cluster number is set to 25. Following Dettmering et al. (2018), by using this number an overall agreement of about 97% can be achieved."

   Against what data this overall agreement was determined? Using data in this paper? If so then it is not a good way for the paper 'progress'. It raises a question that have you tested other numbers of clusters, conducted accuracy analyses, selected 25, and then present accuracy results for this number of clusters.

l. 156: "Moreover, in the quantitative comparison (Sec. 4.2) two additional features are included, which are not used for the UWC:"

   Please explain why they are not used in the UWC.

l. 162: "Figure 2 shows the assignment of 25 clusters to 5 different surface types (compared to 4 in the original UWC approach): undefined, sea ice, thin-ice, lead, and ocean."

   Again here you could describe whether lead class has only open water surface, or if it can have also very thin ice. What is the maximum width of your leads?

Numbers in the color bars in Figure 3 have too small font size.

**4. Results and discussion**

"The first part of this section shows visual comparisons between CryoSat-2 and MODIS as well as Sentinel-1. The second part will then focus on a quantitative analysis of the results."

You could mention that quantitative analysis includes Sections 4.2 and 4.3, if I have understood correctly.

l. 207: "From the altimetry point of view, off-nadir effects may overlay clear leads or thin-ice radar echoes,"

Explain what are these off-nadir effects and how they change waveform parameters.

l. 217: "Qualitatively, the respective classifications appear within expectations"

What are these expectations, how are they quantified?

l. 219: "However, a direct distinction between thicker sea-ice and thin-ice areas is not possible due to the coarse pixel resolution of MODIS in comparison to CryoSat-2."

Is really always the case? A MODIS pixel cannot be covered fully by thin ice or thicker sea ice?

l. 228: "The polynya observed by MODIS appears very bright in the SAR image and is supposedly caused by a rough sea-ice surface due to the presence of frost flowers"

How about possibility that thin ice has rough surface from finger rafting or it has broken to many small floes and frozen again?

Figure 5: What yellow color shows here?

Figure 6: What red color shows here?

l. 241: "In order to account for the much coarser spatial resolution of 12.5km×12.5km for SMOS, CryoSat-2 classifications are aggregated into bins with a length of 12.5km."

How many CS2 surface classification datapoints there are typically in a SMOS SIT pixel?

Note that the SMOS daily SIT chart has 12.5 km pixel, but the original SMOS TB swath data has 35–50 km pixel resolution, and these swath TB's are aggregated to daily TB 12.5 m grid.

In your CS2 vs. SMOS comparison you should also take into account the accuracy of the SMOS SIT: SMOS SIT underestimates sea ice thickness on average by about 50%–60%, and the root mean square difference to validation datasets was 0.31 m:

L. Kaleschke et al., "SMOS sea ice product: Operational application and validation in the Barents Sea marginal ice zone," Remote Sens. Environ., vol. 180, pp. 264–273, Jan. 2016

Did you screened SMOS SIT data using uncertainty and saturation ratio given in the SIT data? "Data with an uncertainty > 1 m or with a saturation ratio near 100% should not be used.", from X. Tian-Kunze. (Nov. 2018). SMOS Sea Ice Thickness ReadMe-First Technical Note (RM-TN).

The used SMOS SIT dataset should be described under Dataset Section.

In Figure 7 the numbers in the color scales are too small.

In Figure 8 the gray bars have poor visibility.

l. 287: "This might be due to an enhanced error budget in the first two thin-ice thickness groups"

What you mean by this enhanced error budget? Please explain in the paper.

l. 290: "The remaining four waveform features, i.e., Wdecay, WfitMAD, LEW, and LWP, feature an apparent correlation, however, it is less clear than the ones we observed with MP, Wwidth, and TES"

Please give numerical correlations in the text.

l. 308: "The flags in the l2i files are based on monthly thresholds for the backscatter coefficient sigma0, the leading-edge width, and pulse peakiness as well as supported by sea-ice-concentration data as sea-ice mask."

Give source of SIC data in the paper.

l. 324: "It should be noted that we define a lead here in the sense of satellite altimetry as an open-water lead, which provides a true sea-surface-height observation without any bias introduced by thin-ice freeboard."

This lead class definition should be earlier in Method Section. Does thin ice really give any freeboard bias in the CS2 data? Lets say 20 cm of thin ice, freeboard is ~1 cm, can this be detected in one CS2 waveform turned into a freeboard? I would assume it is more possible in large averaging of single freeboards, but then you don't have fixed thin ice target for several weeks or days.

l. 333: "Finally, it must be noted that the official definition of the World Meteorological Organisation (WMO) for the term lead includes thin ice with a thickness of up to 30 cm (wmo, 2014)."

This kind of ice type definition should in Introduction.

**5. Summary and outlook**

l. 366: "In contrast, sea-ice classifications also show up in SMOS-derived thin-ice regions."

Here you should investigate the effect of the SMOS SIT accuracy on your comparison results.

---

## Referee Comment (RC2)

Summary

The authors extend an existing unsupervised classification approach for detecting open water targets from radar altimeter waveforms, to include detection of thin-ice. The authors present the classification approach conceptually, apply it to Cryosat-2 L1B Ice Baseline D data waveforms then compare their results to existing thin ice detection techniques from passive microwave radiometry (SMOS sensor) and infrared data (MODIS sensor), as well as Sentinel-1 SAR imagery. A comparison is also made to another Cryosat-2 sea ice product, the AWI Cryosat-2 sea ice product v2.4.  The Laptev Sea is the region of focus due to the occurrence of winter period thin ice areas from flaw leads and coastal polynyas. The classification is applied to Cryosat-2 data taken January through March over the 2011 to 2020 period. Overall, the method is promising for detection of thin-ice areas, as a potential improvement over the ESA Climate Change Initiative (CCI) algorithm described in Paul et al. (2018) which does not include this feature. With a better evaluation in comparison to the prepared MODIS thin-ice thickness data (see major comment below), the utility and limitations of the proposed algorithm will be better understood. The work is clearly justified, and as the authors state, knowledge of thin ice areas is important for sea ice mass balance work, and there are limited datasets for remote study of their properties and coupled processes. It should be a good contribution to TC and of interest to the readership after comments here, and in the other reviews, are adequately addressed.

Major comments:

1. The authors compiled a large number of overlaps between their classification and MODIS TIT information. Despite this, the use of MODIS TIT to understand classifier performance is limited to a brief statement about overall class assignments on lines 267-269. The MODIS TIT data are further used for assessing relationships between ice thickness categories and waveform properties, which adds value to the interpretation of classifier performance. However the paper would be much improved by using the MODIS TIT for more detailed comparisons to the UWC classification outputs, in order to better understand limitations and inform further classifier optimization. The comparison to the CCI data is much more informative.

2. The visual comparisons between the classification and coincident sensor data are helpful, and the authors put together a good summary of relationships between observed conditions in each comparison. The authors should provide a clear rationale for the choices made, in order to add confidence that these are un-biased assessments. It is apparent that the Cryosat-2, MODIS, and Sentinel-1 comparison on 01 March 2018 is chosen due to the short time gap between all three acquisitions. The other comparisons are not as well justified, and it is likely there are several overlaps to choose from.

Minor Comments (by line number):

L4: clarify that it is a Cryosat-2 based classification here

L6: clarify what linear dependency is found (e.g. "between…x and y…") or consider re-wording

L22: delete "in their retrieval capabilities as well as" and replace with "and"

L26: comma after "sensors"

L29: delete "are also prone to"

L36: commas after "density" and "cover"

L38: delete "But even when … classified as sea ice"

L39: change "the later" to "recent" and "over" to "of" (i.e. freeboard *of* ice)

L40: "from" Ku-band radar altimeters

L44: delete "since the range …depth."

L48: delete "on"

L49: delete "short comings and"

L51: Make "However, at a lower spatial resolution …" a new sentence

L58: use "spatial and temporal resolutions" or "spatio-temporal resolution"

L65-68: "This study is structured into the following sections: Section 2 describes the data sets; Section 3 provides details on the unsupervised clustering for CryoSat-2 and the MODIS thin-ice thickness retrieval. Section 4 summarizes and discusses the results and implications on CryoSat-2 surface-type classification, and Section 5 concludes with an outlook."

L70: "The following sub-sections highlight the data sets used …"

L74-75: delete "aiming at monitoring …was placed"

L76: delete "Moreover,"

L79: delete "mainly characterized … ice cover,"

L80: change "showing" to "with"

L85: "This dataset comprises, …"

L86: comma after data, and delete "also", and add "on" after information

L91: delete "As basis" and just use "MODIS" rather than writing it out

L92: delete "MODIS sensors on board the polar orbiting"

L93-94: The MODIS data access information can be moved to the data availability section

L96: delete "In a first step"

L102: Add information about the temporal component, e.g. hourly data nearest to acquisition, daily average, etc.

L110: "and less rough surface under calm, low-wind, conditions." The possibility of wind-roughened polynya and lead should be mentioned since this would also contribute strong backscatter and bright pixel values.

L112: "e.g. on nilas ice, as they…"

L115-116: move the introduction to Sentinel-1 to beginning of the section

L118-120: Since the Sentinel-1 data are used only for visual assessment, the processing steps are likely straightforward and could be described briefly here.

L123-125: delete the first sentence

L127: change "keeping" to "enabling"

L129: delete "used"

L130: clarify what is meant by a Cryosat-2 observation

L131: sentence "The number of useable …" is not necessary because of previous descriptions

L139: delete "has"

L143: put a comma after "clustering" and delete "next classification steps consist of an"

L148: delete "so called" and "followed"

L163" here, and elsewhere, just use the abbreviation after it has been defined (i.e. MP)

L167: nilas

L168: delete "of"

L170: also compare to wind-roughened water

L176: as mentioned above for MP, just use "IST" since it was defined earlier

L191: Change text to "The very high spatial and temporal resolution altimetry …."

Figure 2 caption: use italics for words in quotes (and delete quotation marks)

L199: use "subsets" in place of "zoomed-in snippets"

L200: pluralize to "results"

L207: delete "and, therefore, no truth is available"

L207-209: provide some indication of what is meant by lead or thin-ice surface being very small (how small)

L215: comma after "Northwards"

L216: delete "in ice thickness"

L222: delete "in a quantitative analysis"

L223: change "acquired at" to "from"
L224: delete "from each other"

L229: recommend to change "supposedly" to "likely"

L233: How is thin ice labelled correctly as lead and not thin ice?

L239: change "blended with" to "compared to"

L270: "TIT"

L272: "TIT"

L273: delete "spatiotemporally"

L276: "deviations"

L286: delete "However"

L287: change "this" to "a"

L293: change "Contrariwise" to "In contrast" and use "LEP" only since it is already defined

L304: citation needed

L309: use LEW

L331: delete comma after "both"

L334: "(WMO, 2014)"

L346: delete "in general"

L350: delete "in general" and comma after "both"

---

## Author Comment (AC1)

We thank the Reviewer for the careful and constructive comments. The suggestions and corrections have greatly improved the quality of this manuscript.

**Monitoring Arctic thin ice: A comparison between Cryosat-2 SAR altimetry data and MODIS thermal-infrared imagery**

Felix L. Müller, Stephan Paul, Stefan Hendricks, and Denise Dettmering

**4 July 2022**

**General comments**

The authors have conducted a study which is related to on-going development of sea ice thickness (SIT) retrieval algorithms for CryoSat-2 (CS2) radar altimeter. They modified an existing unsupervised CS2 waveform classification (UWC) method by Müller et al. (2017) to include thin ice class as an additional output (previous sea ice class in now divided into sea ice and thin ice classes). The new UWC was applied to the CS2 data over the Laptev Sea region (as shown in Figure 1) for the winter months Jan-Mar in 2011-2020. The C2 classification results were compared to MODIS thin ice thickness (TIT; max 25 cm) swath charts (in total 161), few Sentinel-1 (S-1) SAR images. SMOS SIT charts, and to ESA CCI CS2 surface classification product which does not have thin ice class. The comparisons are conducted by visual and quantitative analyses. The results suggest (as summarized in Abstract) that there is possibility to either develop simple correction terms for CS2 ranges over thin ice or to directly adjust current retracker algorithms specifically to very thin sea ice. Further, the new UWC can rather reliably discriminate between sea ice, open water leads, and thin ice within recently refrozen leads or mere areas of thin sea ice. The new UWC surface type classification should be better for freeboard and SIT retrieval than the current ESA CCI CS2 product, e.g. the new UWC gives much less unknown surface type classifications. It may be possible, after further developments, to estimate SIT of thin ice based on the CS2 waveform data (freeboard methods lacks sensitivity to very small freeeboards of thin ice).

I think the study set up with data acquisitions and data processing is sound, as are methods for the data analyses. However, I think there are some room for improvements, and in the following I have some general questions and comments, followed by miscellaneous specific ones.

First, the you should clearly define in the Abstract and Introduction what you mean by 'thin ice', what is the maximum assumed thickness for it? I think it is not good to use term 'young ice' in this context, as in the WMO sea ice nomenclature it is a sea ice class with 10-30 cm thickness.

In accordance with a comment by Reviewer 2 we changed the abstract (L6) to read:
*"Here, strong linear dependencies are found between binned thin-ice thicknesses up to 25cm thickness from MODIS and the CryoSat-2 waveform shape parameters that show the possibility to either develop simple correction terms for altimeter ranges over thin ice or to directly adjust current retracker algorithms specifically to very thin sea ice."*

Furthermore, we added also in accordance with a reply to Reviewer 2 L38 to read:

*"For correct sea-ice classifications, the small freeboard values of thin ice (here defined as sea ice with a thickness up to 25 cm) are often lower than the precision of even the later synthetic aperture radar (SAR) altimeter sensors."*

*We agree with the misleading use of 'young ice' and changed all occurrences to 'thin ice' used with the provided range description above.*

All the used datasets should be presented under Dataset Section, this is not the case now for SMOS SIT data and the ESA CCI product, e.g. the ESA CCI product is now described in Section 4.3.

*It is true that the SMOS dataset is not mentioned in the datasets. We would like to focus intentionally on Cryosat-2, MODIS and Sentinel-1, since these data were mainly used. In the case of SMOS, a ready-made data product was used. The dataset was taken unchanged as a basis for comparison and is therefore well documented and referenced, whereas in the case of Cryosat-2, MODIS and Sentinel-1 observations had to go through various pre-processing steps.*

*The sources of all datasets (including SMOS) are listed in "data availability" at the end of the paper.*

The discussion on thin ice remote sensing methods in Introduction is quite brief and should be expanded. This can focus on altimeter data based studies and methods, but it is good to discuss also other ones, e.g. discussion on high frequency radiometer data based thin ice detection is very brief, there have be many studies on recent years, see e.g.:

*K. I. Ohshima, S. Nihashi, and K. Iwamoto, "Global view of sea-ice production in polynyas and its linkage to dense/bottom water formation," Geosci. Lett., vol. 3, no. 1, p. 13, 2016.*

*K. Nakata, K. I. Ohshima, and S. Nihashi, "Estimation of thin-ice thickness and discrimination of ice type from AMSR-E passive microwave data," IEEE Trans. Geosci. Remote Sens., vol. 57, no. 1, pp. 263–276, Aug. 2019.*

*K. Nakata, K. I. Ohshima, and S. Nihashi, "Mapping of active frazil for Antarctic coastal polynyas, with an estimation of sea-ice production," Geophys. Res. Lett., vol. 48, no. 6, Mar. 2021, Art. no. e2020GL091353.*

First study about frazil ice monitoring.

*We thank the reviewer for her/his suggestions. Our clear intention is to focus on the altimetry part as was also pointed out by the reviewer and, therefore, limit our discussion of other approaches. Nevertheless, we already included a wide variety of other studies using several different sensor types in L19/20. Moreover, since we focus primarily on the Arctic Ocean, we decided against including Antarctic example studies.*

It is nice that the authors are also using S-1 SAR imagery as reference data, but only with visual analysis. Currently, there are many automatic S-1 SAR sea ice classification algorithms available, e.g. this one especially for lead detection:

*Murashkin et al., Method for detection of leads from Sentinel-1 SAR images, Annals of Glaciology (2018), doi: 10.1017/aog.2018.6*

This algorithm was developed at UB and AWI, so maybe the authors have access to it? At least lead detection by SAR should be briefly discussed. See also S-1 SAR sea ice classification studies, e.g.:

Park, J.-W., Korosov, A. A., Babiker, M., Won, J.-S., Hansen, M. W., and Kim, H.-C.: Classification of sea ice types in Sentinel-1 synthetic aperture radar images, The Cryosphere, 14, 2629–2645, https://doi.org/10.5194/tc-14-2629-2020, 2020.

Boulze, H.; Korosov, A.; Brajard, J. Classification of Sea Ice Types in Sentinel-1 SAR Data Using Convolutional Neural Networks. Remote Sens. 2020, 12, 2165. https://doi.org/10.3390/rs12132165

Previous altimeter lead/thin ice detection studies should also be discussed in more detail, their methods and results summarized, and later to compare the methods/results of this study to the previous ones, maybe it would be good to add Discussion Section. Some previous related studies:

Lee et al., Arctic lead detection using a waveform mixture algorithm from CryoSat-2 data, The Cryosphere, 12, 1665–1679, 2018, https://doi.org/10.5194/tc-12-1665-2018

N. Longépé et al., "Comparative Evaluation of Sea Ice Lead Detection Based on SAR Imagery and Altimeter Data," in IEEE Transactions on Geoscience and Remote Sensing, vol. 57, no. 6, pp. 4050-4061, June 2019, doi: 10.1109/TGRS.2018.2889519.

Wernecke, A. and Kaleschke, L.: Lead detection in Arctic sea ice from CryoSat-2: quality assessment, lead area fraction and width distribution, The Cryosphere, 9, 1955–1968, https://doi.org/10.5194/tc-9-1955-2015, 2015

*In response to this comment, we have included some of the studies mentioned in the introduction. The*

*manuscript has been supplemented accordingly:*

*Numerous studies exist on the automatic detection of leads. These use altimeter data (Lee et al, Wernecke et al, Dettmering et al) or SAR images (Park). [...] However, none of the existing studies has yet attempted to detect thin-ice using satellite altimetry data.*

I guess their relevance to the study depends how 'leads' are defined in them, are they only open water covered or also may have thin ice cover? In your CS2 UWC classification how you do define differences between classes 'lead' and 'thin ice'? Your 'lead' has only open water surface, or could it also have few cm of thin ice coverage?

In our case, leads are open water areas without a thin ice cover. We note that some definitions of leads may also include the presence of thin ice but for the purpose of satellite altimetry, a lead is a surface that allows us to observe sea surface height directly.

You could shortly summarize in Introduction the comparison results from:

Dettmering, D., Wynne, A., Müller, F. L., Passaro, M., and Seitz, F.: Lead Detection in Polar Oceans—A Comparison of Different Classification Methods for Cryosat-2 SAR Data, Remote Sensing, 10, https://doi.org/10.3390/rs10081190, 2018.

We added this paper in the introduction (see above). However, we decided against giving a more detailed summary since this is outside the scope of this paper.

It is understandable to you have a limited study region, and the Laptev Sea with recurring polynyas is good for it, but it would have been nice to see how your new UWC performs over some MIZ, like Barents Sea where large areas of thin ice may occur at the ice edge.

We thank the reviewer for this suggestion. Indeed, it's interesting to extend the thin ice investigations to MIZ. We take this as a suggestion for further analyses and add a sentence to the outlook section.

*Moreover, to enable further Arctic climate-relevant investigations using altimetry data the presented classification of thin-ice can be extended to the entire Arctic **Ocean including its peripheral seas and marginal ice zone** to produce maps of altimetry-derived thin-ice coverage.*

AWI conducts HEM flights over the western Arctic, would it be possible to include also HEM data to your study? It would give accurate data on the leads and thin ice, although temporal and spatial matching with CS2 tracks is likely a problem. You could discuss about this in Introduction, even in case the HEM data are not added.

Airborne EM data sets do not have the extent, coverage or precision to be useful for our study. This comes on top of the reviewers correct assessment that colocation alone will be difficult. Our focus therefore remains on the MODIS data as reference for thin ice thicknesses.

Did you investigate also using Sentinel-2 imagery as lead/thin ice reference data? It has been used in some sea ice studies, e.g.

No, currently we're not working with Sentinel-2 scenes due to the limited availability during the winter months and the fact that we are focusing on MODIS-derived thin-ice thickness. Moreover, the swath width of Sentinel-2 is very limited compared to Sentinel-1, which makes it quite difficult to find overlaps close in time. But of course, Sentinel-2 and the Landsat family provide highly resolved images of the current sea ice condition, which can be used in future investigations as a data source for visual comparisons.

Muchow, M., Schmitt, A. U., and Kaleschke, L.: A lead-width distribution for Antarctic sea ice: a case study for the Weddell Sea with high-resolution Sentinel-2 images, The Cryosphere, 15, 4527–4537, https://doi.org/10.5194/tc-15-4527-2021, 2021.

Petty, et al. (2021). Assessment of ICESat-2 sea ice surface classification with Sentinel-2 imagery: Implications for freeboard and new estimates of lead and floe geometry. Earth and Space Science, 8, e2020EA001491. https://doi. org/10.1029/2020EA001491

Why did you not use the latest CS2 L1B baseline-E data? https://earth.esa.int/eogateway/news/new ice-baseline-e-and-near-real-time-processors

We used the CS2 L1B D baseline because it is fully available for the entire mission duration of Cryosat-2 since 2010. Baseline E is the latest reprocessing, but only available for a few months and years and not complete.

I have following questions on the MODIS data in Section 2.2 and MODIS TIT calculation in Section 3.2:

How cloud masking is conducted?

Only manual visual screening was conducted as the MODIS cloud mask tends to eliminate especially very thin ice areas (see. Paul and Huntemann, 2021; The Cryosphere). However, a similar product as detailed in the article is not yet available for the Arctic.

Are both daytime and nighttime MODIS images used?

Only night-time data is used. We clarified that in the text.

Why did you not use the MOD29 product which has cloud masked IST?

Following our reasoning above, this would potentially lead to thin-ice exclusions from our analysis due to the false-positive classifications of thin ice as cloud in the MODIS cloud mask product. Therefore, we calculated our own IST following the algorithm/user documents of the MOD29 product.

What ERA5 fields were used, hourly? If so then reference for this is:

We thank the reviewer for pointing this out to us. Also with respect to a comment by Reviewer 2 we changed this line to clarify our use of hourly data and exchanged the provided reference.

*"These fields comprise the 2 m air temperature, the 10 m wind-speed components, the mean sea-level pressure, and the 2 m dew-point temperature in hourly resolution."*

H. Hersbach et al. (2018). ERA5 Hourly Data on Single Levels From 1979 to Present. Copernicus Climate Change Service (C3S) Climate Data Store (CDS). Accessed: X. [Online]. Available: https://doi.org/10.24381/cds.adbb2d47

Why did you not use ERA5 mean surface downward long-wave radiation flux? How it was now obtained?

As outlined in the text, details to the retrieval algorithm are provided in Paul et al. (2015) to reduce redundancies. The retrieval of the surface downward long-wave radiation flux follows our established approach using an improved parametrization by Jin et al. (2006). Switching this up would the material for a dedicated future study and likely rise additional questions for this study.

*Jin, X., Barber, D., and Papakyriakou, T.: A new clear-sky downward longwave radiative flux parameterization for Arctic areas based on rawinsonde data, J. Geophys. Res., 111, D24104, doi:10.1029/2005JD007039, 2006.*

Is snow on thin ice neglected in the TIT calculation?

Yes. This is one assumption in general for the retrieval of thin-ice thickness from MODIS in our and also other approaches and also outlined in, e.g., Paul et al (2015).

MODIS TIT accuracy is given according to Adams et al. (2012), but is your TIT calculated exactly as in this paper? If not then their accuracy figure may not be valid for your TIT.

We thank the reviewer for pointing this out to us. The reviewer is correct in his/her implied assumption that there were changes made between the study by Adams et al. (2012) and ours. Primarily the used atmospheric reanalysis changed from NCEP2 (Adams et al.) to ERA5 (this study). The provided uncertainty shall remain as an indicator for an uncertainty estimate, however, due to the improvements with reanalysis data, we would assume an overall better accuracy for the

MODIS thin-ice retrieval compared to Adams et al.. Nonetheless, this would also be subject to a dedicated study and can not be provided within the scope of this study.

We adjusted the text in the manuscript to read:

*"While no state-of-the-art uncertainty analysis is available for our combination of MODIS thin-ice retrieval with ERA5 atmospheric reanalysis, Adams et al. (2012) state an average uncertainty of ±4.7 cm for ice thicknesses between 0.0 and 0.2 m using NCEP2 reanalysis (Kalnay et al., 1996), substantially increasing for larger thickness ranges"*

Your description of new UWC in Section 3.1 does not include any information on the weather conditions. Your study was conducted in winter conditions, but still is there any dependence on resulting clusters on the air temperature? Thin ice surface condition: slushy, dry, frost flowers etc., depend on the weather conditions. What was the air temp range in your study? Any above zero cases?

From ERA5 2m air temperature data, the average temperatures for the study period and region is always well below freezing point as one would expect in winter (about 253K depending on the exact location; a). However, there are rare occasions of above freezing-point temperatures (about 1.6%; b) especially over land, but also near the coast over sea-ice/fast-ice areas. While certainly the surface conditions change under these conditions and impact the received returns for CryoSat-2, we consider the overall impact on the UWC negligible.

[Figure]

You should summarize in Section 3.1 rules how the 25 clusters are classified to different surface classes, e.g. with table or flowchart. You could also show statistics and pdf's of some features for different classes. In general, your description how clusters are classified is now very rough.

We understand the reviewer's suggestion, however, all processing and assignment steps are included and described in detail in various publications and reports (see Müller et al., 2017, Dettmering et. al, 2018, Passaro et al. 2020). This study is more about the application and extension of the already developed unsupervised classification approach than about the development of the classification itself, which has already been described with validation procedures in publications listed above.

On backscattering mechanism discussion in lines 230-231 and 281-283: you should include discussion on the coherent vs. non-coherent scattering for radar altimeter. The coherent scattering is very high, increasing roughness decreases it quickly and non-coherent scattering due to roughness increases.

We have added a description of the backscatter mechanisms for thin ice that in our view cause the increase in waveform width and decrease in peak power. We also added a reference (Landy et al., 2020) where the reader can find more information on coherent and incoherent backscatter mechanism for sea ice radar altimetry.

> *[...] We note here, that this change in waveform properties can be caused by a change roughness at radar wavelength scale or large scales, e.g. by ice deformation. It is unlikely that larger areas covered solely by thin sheet ice have significant surface height variability nor a considerable snow layer. But is has been shown by Landy et al., 2020 that the increase of radar scale roughness can significantly affect CryoSat-2 waveform shape even for*

*comparably flat surfaces when incoherent backscatter from radar scale roughness is taken into account. This effect may be so pronounced that several waveforms are assigned to the sea ice category in an area where the SAR image indicates the presence of thin ice only.*

In Section 4.2 Quantitative analysis the CS2 vs. MODIS comparison results should be presented also in a table/matrix format. Give also the accuracies for CS2 thin ice and thick ice classes against the MODIS TIT charts, and e.g. commission error or False Alarm Rate: percentage of MODIS thin ice pixels falsely classified as thick ice out of all MODIS thin ice pixels.

The matrix is shown below with a categorization of MODIS thin ice thickness below or above 20 cm. There are indeed a considerable amount of sea ice classifications in areas indicated as thin ice by MODIS. But we need to consider that with CryoSat-2 and MODIS we are comparing two sensors with different resolution. An waveform classified as sea ice in a thin ice MODIS pixel is potentially a correct classification if there has been small sea ice parts present, or the situation changed between the MODIS and CryoSat-2 acquisitions. Specifying a commission error of False Alarm rate would neglect this uncertainty.

We however decided to not add the matrix to the manuscript.

| Classes MODIS TIT | MODIS TIT <=0.2 m | 0.2 m < MODIS TIT <0.25 m | ∑ |
|---|---|---|---|
| Ocean | 0 | 0 | 0 |
| Lead/polynya | 0.18 | 0.02 | 0.20 |
| Thin ice | 0.20 | 0.10 | 0.30 |
| Sea ice | 0.31 | 0.14 | 0.45 |
| undefined | 0.04 | 0 | 0.04 |
| ∑ | 0.73 | 0.26 | ~1 |

Total sum observations: 213540 (only MODIS overlaps).

It would be interesting to also see how CS2 thin ice classification works as a function of MODIS TIT, a similar investigation is now conducted for CS2 waveform features.

This comment leads in the direction of the table shown in the comment above. In the present investigation we're focusing on recently refrozen leads, with only a limited spatial extent compared to the spatial resolution of one MODIS pixel. Due to this resolution difference, there cannot be a direct pointwise and fair comparison between TIT and the UWC results.
Moreover, the assignment process of the classes to surface types follows the principle of being able to recognize thin ice as reliably as possible. This leads to the fact that the percentage in the range <=20 cm between "Sea ice" and "Thin ice" is very similar or the sea ice detections predominate.

In Summary and outlook Section: "This fact brings the future task to check further waveform clusters regarding possible thin-ice detections."

What this exactly means, that you are not sure yet what surface classes all your 25 clusters really represent? If not then you should conduct further studies already in this paper.

This sentence is meant that we can detect the prominent waveforms representing thin ice as such. The aim is to safely classify thin ice. However, we do not know to what extent influences of wind and waves during thin ice conditions can possibly extend to other waveform clusters.

This is a suggestion for a possible future investigation, but goes beyond the scope of the current investigation. It is likely that the current investigation would have to be extended to the entire Arctic to be able to use or observe sufficient thin-ice conditions in this way. The sentence stands for the

still open questions and possibilities which the classification might offers.

**Spesific comments**

Abstract

"The ESA Climate Change Initiative's (CCI) surface-type classification"

> You should mention that this is also based on the CS2 data, and that it does not have thin ice class.

Agreed and changed.

"[...] the basic ESA Climate Change Initiative's (CCI) CryoSat-2 surface-type classification with classes 'sea ice', 'lead', and 'unknown' [...]"

**1. Introduction**

Page 1, line 21: "However, the resulting data sets are generally bound to an upper sea-ice thickness limit in their retrieval capabilities as well as methodological limitations (e.g. cloud cover presence when using thermal-infrared data"

> It would be good to mention here also the effect of snow cover.

We added "snow cover" as an example.

l. 27: "However, studies suggest a higher uncertainty towards thinner sea-ice (Ricker et al.,

> 2017)." Please give the current 'accurate' SIT retrieval range of CS2.

Sea ice thickness uncertainty in absolute terms is lower for thin ice than for thicker sea ice, since the considerable impact of sea ice density uncertainty is almost linear related to freeboard. For thin ice, the uncertainty component of the snow mass is more important. This leads to a significant increase in relative uncertainty though, as described in Ricker et al., (2017). The exact limit of 'accurate SIT retrieval range' does depend on the specific definition of accuracy and the exact choice of radar altimeter processing and auxiliary data sets. No universal number can or should be given. This study also has a focus on surface-type classification and not freeboard retrieval or freeboard to thickness conversion. Interested readers are directed to the respective Ricker et al. (2017) paper.

In the CS2 SIT retrieval discussion you could also mention the effect of snow salinity, decreases penetration depth:

Nandan, V., et al. (2017). Effect of snow salinity on CryoSat-2 Arctic first-year sea ice freeboard measurements. Geophysical Research Letters, 44. https://doi.org/10.1002/2017GL074506

We thank the reviewer for pointing this out to us. This is a general issue on the altimetry retrieval but not necessarily limited to thin ice which we would like to focus on in this section of the manuscript.

l. 48: "While information on the presence of thin-ice areas is important for our understanding on sea-ice mass balance changes, there is currently only a single operational thin-ice data product available due to the above-mentioned short-comings and limitations."

> JAXA has AMSR2 based operational research product "Detection of thin sea ice", see https://suzaku.eorc.jaxa.jp/GCOM_W/data/data_w_product-3.html

We thank the reviewer for pointing this out to us. We were not aware of this product. However, from what information is provided by the given link, this product covers only 3 marginal Arctic seas and not the Arctic basin what we would aim for in the future using a thin-ice aware CryoSat-2 data product.

To highlight this, we changed the above line to read:

*"[...] there is currently only a single operational Arctic-wide thin-ice data product available [...]"*

l. 52: "This method is limited for thicker sea ice and thus data fusion of SMOS and CryoSat-2 using Optimal Interpolation"

    Give the maximum SIT for an 'accurate' SMOS SIT estimation, in papers by Kaleschke and Tian-Kunze et al.

In Ricker et al. (2017), the authors draw the line at 1m of ice thickness as there the SMOS retrieval loses its sensitivity while the opposite is true for CryoSat-2.

We added this information to the mentioned line.

Ricker, R., Hendricks, S., Kaleschke, L., Tian-Kunze, X., King, J., and Haas, C.: A weekly Arctic sea-ice thickness data record from merged CryoSat-2 and SMOS satellite data, The Cryosphere, 11, 1607–1623, https://doi.org/10.5194/tc-11-1607-2017, 2017.

l. 56: "In this study, the authors utilize Delay-Doppler radar altimeter echoes from ESA's Earth Explorer mission CryoSat-2 in combination with the capabilities of NASA's Moderate Resolution Imaging Spectroradiometer (MODIS) to monitor thin ice with a high spatial-temporal resolution."

    To my understanding MODIS TIT data is used for comparison, and not in combination with CS2 for thin ice monitoring.

We thank the reviewer again for pointing this out to us as it was phrased indeed misleadingly. We changed it to read:

*"[...] Moderate Resolution Imaging Spectroradiometer (MODIS) to enable future monitoring of thin ice with a high spatial-temporal resolution."*

l. 62: "flaw and coastal polynyas"

    Describe shortly what these are; good general info for readers.

We changed the line to read:

*"[...] Arctic Laptev-Sea region featuring frequent occurrences of flaw and coastal polynyas, i.e., polynyas bound to fast ice or the coast, respectively."*

**2. Dataset**

Or Datasets and their processing?

We thank the reviewer for this suggestion and change "Dataset" to "Data sets and pre-processing".

l. 87: "ESA Cryosat-2 Product Handbook (https://earth.esa.int/eogateway/documents/20142/37627/CryoSat-Baseline-D-Product-Handbook.pdf/c76df710-2a5c-c8b8-00c1-13c8db0e9f51,"

    Make this as reference.

We treated the link of the handbook like it was treated in Meloni et. al, 2020, https://tc.copernicus.org/articles/14/1889/2020/. However, we also think it's better to have it in the bibliography. We changed that.

l. 97: "2009). Subsequently, the sea-ice-surface temperature (IST) was computed following Riggs and Hall (2015)."

Original reference for MODIS IST is:

Hall, D.; Key, J.; Casey, K.; Riggs, G.; Cavalieri, D. Sea ice surface temperature product from MODIS. IEEE Trans. Geosci. Remote Sens. 2004, 42, 1076–1087.

We appreciate the hint but we would stick with the given reference to the MODIS Sea Ice Products User Guide to Collection 6 which is what we actually used to calculate the IST with the state-of-the-art parameters used by the NSIDC.

In your discussion of S-1 SAR backscatter signatures for leads a good reference is:

Murashkin et al., Method for detection of leads from Sentinel-1 SAR images, Annals of Glaciology (2018), doi: 10.1017/aog.2018.6 which discusses about dark and bright leads. This paper could also be as your basis for visual lead identification in the SAR imagery (if you keep doing so, and not applying some automatic method).

We thank the reviewer for this interesting study. We added the reference to chapter 2.3.

Give the total number of S-1 SAR images used in your study.

We are not quite sure what the deeper meaning of this comment is. The S-1 SAR images are used for visual comparisons only; no systematic studies were performed as we focus on analyses using MODIS-derived thin-ice thickness.

The SAR images shown in the manuscript are the best way to demonstrate the functionality of unsupervised classification and thin ice detection. The number of images generally depends on the availability of the two missions and the number of comparison points. Often there is no overlap in the image area or only minimal common overlap areas. At the same time, of course, the selected study period and area forms a further limitation.

In terms of temporal and spatial constraints, 508 S-1A/B images are theoretically available. With the addition of a maximum observation gap of 30 minutes, 52 scenes remain, which were filtered again with respect to vividness and available overlap points. In the end, this results in a number of 8 examples.

We added a sentence to manuscript and changed text parts of section 2.3.

*For an improved understanding of the CryoSat-2/MODIS comparison the authors utilize images from the side-looking ESA Copernicus C-Band SAR missions Sentinel-1A and B (S1-A/B). For a better understanding of the CryoSat-2/MODIS comparison, images from the side-looking ESA Copernicus C-Band SAR missions Sentinel-1A and B (S1-A/B) are used for a visual comparison since they are unaffected by cloud cover. All acquired S1-A/B scenes were used for additional visual comparison as they are unaffected by cloud cover. [...] In terms of study area and period, the S-1A/B image pool represents a theoretical size of approximately 500 scenes.*

Furthermore, we add a sentence to chapter 3 (methods):

*After setting a maximum permissible acquisition time difference between Cryosat-2 and Sentinel-1A/B, the number of possible SAR images is reduced to 52. Other limiting factors concern the actual number of overlap points in the image area. The images shown in Section 4.1 correspond to examples with a very good overlap and different surface conditions.*

l. 117: "The images are ground-range detected and show a pixel resolution of 40m and a swath width of 400km."

The pixel size is 40 by 40 m, and spatial resolution is close 100 m depending on range.

We added the information.

l. 118: "The images are processed using the SNAP - ESA Sentinel Application Platform v8.0, (http://step.esa.int) following the processing steps described in Müller et al. (2017) and Passaro et al. (2018b), but with an additional speckle filtering."

Please summarize these processing steps.

The processing steps are very well summarized and described in the provided references (Müller et al., 2017 and Passaro et al., 2018b). We want to keep the manuscript short with focus on the shown applications and investigations. Nevertheless, we add a sentence to chapter 2.3.

*[...] Briefly summarized, the main processing steps follow standard routines (e.g. radiometric calibration, speckle filtering, map projections etc.).*

**3. Methods**

l. 129: "The MODIS comparison database used includes 161 scenes, which corresponds to an altimetry dataset of about 21300 Cryosat-2 observations (MODIS scenes and their time difference from the Cryosat-2 observations are given out in a list upon request)."

What is the spatial resolution of these CS-2 observations? Just the area for one CS2 waveform? And there can be several waveforms within one 1 km MODIS IST pixel?

Since satellite altimetry is a height point measurement method, there is no direct area resolution as for example with side-looking SAR sensors. We used the 20Hz CS2 SAR dataset, which corresponds to an observation interval of about 300 meters, depending on latitude and surface conditions. The across-track footprint size is given by 1.65km. This means there might be up to 3 CS2 observations in one MODIS TIT pixel.

We add a sentence in chapter 4.1 (visual comparison):

*[...] Which leads to the fact that on one MODIS TIT pixel can fall up to 3 Cryosat-2 observations [...]*

l. 137: spell out LRM acronym.

We deleted the acronym "LRM" since it is more confusing in this context and changed the text to:

*Müller et al. 2017 applied it to the non-SAR, conventional altimeter missions Envisat and SARAL*

l. 139: "However, none of these studies has classified thin-ice, since altimeter waveforms generated by this surface type are quite similar to open water returns."

Can you at this point summarize the difference between the open water and thin ice backscatter/waveforms based on earlier studies?

To our knowledge, there are no previous studies describing the differences between the waveforms of open water and thin ice. An explanation by us can be found in L168.

l. 142: "After clustering next classification steps consist of an assignment of the clustered waveforms as well as remaining waveforms to certain surface types (e.g. ocean, lead/polynya or sea-ice conditions)."

How this is conducted? I would assume some automatic method has been developed, please describe it.

This step is a mixture of applying physical knowledge about the waveform scatter characteristics by the different surface types and a statistical analysis of the mean feature values (see Figure 2).

The selection process is well documented in Müller et al. (2017), Passaro et al. (2020), and Dettmering et al. (2018), to which we would like to refer. However, we have added a sentence to the manuscript, chapter 3.1.

*The assignment of the waveform clusters to the different surface types is mainly based on the use of background knowledge about the physical backscattering properties of the individual surface types and statistical relationships (see Figure 2). More detailed explanations of the cluster assignment can be found in Müller et al., 2017.*

l. 144: "In the present investigation, the cluster number is set to 25. Following Dettmering et al. (2018), by using this number an overall agreement of about 97% can be achieved."

Against what data this overall agreement was determined? Using data in this paper? If so then it is not a good way for the paper 'progress'. It raises a question that have you tested other numbers of clusters, conducted accuracy analyses, selected 25, and then present accuracy results for this number of clusters.

The number of clusters is taken from the Dettmering et al. study. Also the percentage of overall agreement stems from that study and the data used there. We agree that this might be misleading and the sentence has been changed to:

*In the present investigation, the cluster number is set to 25. This number was found to produce optimal results in Dettmering et al. (2018) and let there to an overall agreement of about 97%.*

l. 156: "Moreover, in the quantitative comparison (Sec. 4.2) two additional features are included, which are not used for the UWC:"

Please explain why they are not used in the UWC.

The two additional features are part of a classification developed under the ESA CCI (described in Paul et al. 2018), which was developed at a different time and with a different approach. However, we would like to investigate to what extent these two additional features behave with respect to different thin ice thicknesses. Therefore, we decided to include LEW and LEP in the study even though there is no direct connection to the UWC.

Generally speaking, these features can theoretically also be part of the UWC, but that is not the focus here.

We have rephrased the sentence and added a comment in the introduction referring to the ESA CCI classification and the UWC statistical analysis in Section 4.3.

l. 162: "Figure 2 shows the assignment of 25 clusters to 5 different surface types (compared to 4 in the original UWC approach): undefined, sea ice, thin-ice, lead, and ocean."

Again here you could describe whether lead class has only open water surface, or if it can have also very thin ice. What is the maximum width of your leads?

We are not quite sure if we have understood this comment correctly. Basically, there is no maximum width of leads, because we are looking for open water spots in the sea ice with CS2. Therefore, there can be only one lead classification or several depending on the width of the lead. If a very narrow lead lies between observation points, it will not be detected. If a lead is very wide, either many lead classifications correspond to it or it will be detected as open ocean at some point. When a lead surface is classified as thin ice, we do not recognize the lead itself but the thin ice surface even if it has formed into a lead.

Numbers in the color bars in Figure 3 have too small font size.

We agree to this comment and changed that.

**4. Results and discussion**

"The first part of this section shows visual comparisons between CryoSat-2 and MODIS as well as Sentinel-1. The second part will then focus on a quantitative analysis of the results."

You could mention that quantitative analysis includes Sections 4.2 and 4.3, if I have understood correctly.

We agree to the reviewer and accept the suggestion. We added some text to Section 4.

*In the last part, Section 4.3 briefly evaluates the representation of thin-ice-class waveforms in other*

*CryoSat-2 based products (Paul et al., 2018) compared to the unsupervised classification approach.*

l. 207: "From the altimetry point of view, off-nadir effects may overlay clear leads or thin-ice radar echoes,"

Explain what these off-nadir effects are and how they change waveform parameters.

Off-nadir effects are a well-known issue in satellite altimetry. Off-nadir effects can occur in the inland area, but also in the sea-ice domain. An off nadir effect occurs when a strong backscatterer (e.g. specular lead) that is not in the nadir of an altimeter, but rather off center or at the edge of the footprint dominates the reflective signal. These effects can lead to a wrong height determination.

We added some text.

*From the altimetry point of view, off-nadir effects, such as when a dominant and specular lead is not in the nadir direction, may overlay clear leads or thin-ice radar echoes, which hence prevents a clear identification, in particular if the lead or thin-ice surface is very small. These off-nadir effects can become noticeable in the waveform by the appearance of further dominant peaks in the backscatter signal, which later can lead to deviations in the height determination if a retracking algorithm specially modified for this problem is not used (see e.g. Quartly et al., 2019).*

l. 217: "Qualitatively, the respective classifications appear within expectations" What are these expectations, how are they quantified?

These expectations are outlined in the next lines (218/219) and are not quantified as this is just a qualitative comparison/example to provide an easy entry to both data sets for the readers.

l. 219: "However, a direct distinction between thicker sea-ice and thin-ice areas is not possible due to the coarse pixel resolution of MODIS in comparison to CryoSat-2."

Is really always the case? A MODIS pixel cannot be covered fully by thin ice or thicker sea ice?

The reviewer is likely correct in his assumption and we clarified our text to read:

*"However, in most cases a direct distinction between thicker sea-ice and thin-ice areas is not possible due to the coarse pixel resolution of MODIS and the variability of ice thickness within a MODIS pixel in comparison to CryoSat-2."*

l. 228: "The polynya observed by MODIS appears very bright in the SAR image and is supposedly caused by a rough sea-ice surface due to the presence of frost flowers"

How about possibility that thin ice has rough surface from finger rafting or it has broken to many small floes and frozen again?

We assume the suggestion by the reviewer could also cause higher backscatter returns. We modified our test to state frost flowers as just one example.

Figure 5: What yellow color shows here?

We add an explanation to the caption.

Figure 6: What red color shows here?

We add an explanation to the caption.

l. 241: "In order to account for the much coarser spatial resolution of 12.5km×12.5km for SMOS, CryoSat-2 classifications are aggregated into bins with a length of 12.5km."

How many CS2 surface classification datapoints there are typically in a SMOS SIT pixel?

This is not a fixed number, but theoretically one SMOS pixel can correspond to up to 50 CS2

observations. In fact, in the examples shown (Figure 7), 41 - 42 CryoSat-2 observations correspond to one SMOS pixel.

We added this information to the text.

Note that the SMOS daily SIT chart has 12.5 km pixel, but the original SMOS TB swath data has 35–50 km pixel resolution, and these swath TB's are aggregated to daily TB 12.5 m grid. In your CS2 vs. SMOS comparison you should also take into account the accuracy of the SMOS SIT: SMOS SIT underestimates sea ice thickness on average by about 50%–60%, and the root mean square difference to validation datasets was 0.31 m:

> L. Kaleschke et al., "SMOS sea ice product: Operational application and validation in the Barents Sea marginal ice zone," Remote Sens. Environ., vol. 180, pp. 264–273, Jan. 2016

> Did you screened SMOS SIT data using uncertainty and saturation ratio given in the SIT data? "Data with an uncertainty > 1 m or with a saturation ratio near 100% should not be used.", from X. Tian-Kunze. (Nov. 2018). SMOS Sea Ice Thickness ReadMe-First Technical Note (RM-TN).

No we used the SMOS SIT data as provided and just screened it for large polynya openings to increase the number of comparisons.

The used SMOS SIT dataset should be described under Dataset Section.

Please see our answer to this comment above.

In Figure 7 the numbers in the color scales are too small.

We updated the plot.

In Figure 8 the gray bars have poor visibility.

The gray bars should be in the background. We think it's important to provide a standard deviation, however this standard deviation should not be the focus of that plot. Nevertheless, we updated the plot.

l. 287: "This might be due to an enhanced error budget in the first two thin-ice thickness

> groups" What you mean by this enhanced error budget? Please explain in the paper.

We rephrased this line to read:

"This might be due to a larger uncertainty associated with the first two thin-ice thickness bins

> and their in general lower occurrence rates compared the thicker thin-ice bins."

l. 290: "The remaining four waveform features, i.e., Wdecay, WfitMAD, LEW, and LWP, feature an apparent correlation, however, it is less clear than the ones we observed with MP, Wwidth, and TES"

Please give numerical correlations in the text.

We follow the reviewer's suggestion and provided numerical (Pearson) correlations. Please find a table in the Appendix.

l. 308: "The flags in the l2i files are based on monthly thresholds for the backscatter coefficient sigma0, the leading-edge width, and pulse peakiness as well as supported by sea-ice-concentration data as sea-ice mask."

> Give source of SIC data in the paper.

We have added the sea ice concentration source products to the manuscript.

l. 324: "It should be noted that we define a lead here in the sense of satellite altimetry as an open

water lead, which provides a true sea-surface-height observation without any bias introduced by thin-ice freeboard."

This lead class definition should be earlier in Method Section. Does thin ice really give any freeboard bias in the CS2 data? Let's say 20 cm of thin ice, freeboard is ~1 cm, can this be detected in one CS2 waveform turned into a freeboard? I would assume it is more possible in large averaging of single freeboards, but then you don't have fixed thin ice target for several weeks or days.

What we mean here is that thin ice freeboard constitute a systematic bias to sea surface height estimates when the respective surface is classified as an open water lead. Sea ice thickness information from radar altimetry is mostly aggregated over a large number of orbits before the scientific analysis. And in winter time it is highly likely that leads will be covered by some type of thin ice and thus will be biased high on average.

We added and modified the following text in the introduction:

*[...] A lead in the context of radar altimetry is an ice-free opening surrounded by sea ice that allows to directly measure sea surface height within the ice cover. Continuous sea surface height observations over open ocean and at discrete lead locations are then interpolated along the track and one can subsequently calculate the sea-ice freeboard (i.e., the instantenous height differences between the sea-ice surface and the ocean surface).*

But the reviewer is correct in the assertation that range accuracy in the order of 1 cm is out of scope for individual waveforms.

l. 333: "Finally, it must be noted that the official definition of the World Meteorological Organisation (WMO) for the term lead includes thin ice with a thickness of up to 30 cm (wmo, 2014)."

This kind of ice type definition should in Introduction.

Definition has been added to introduction

**5. Summary and outlook**

l. 366: "In contrast, sea-ice classifications also show up in SMOS-derived thin-ice regions."

Here you should investigate the effect of the SMOS SIT accuracy on your comparison

results.

We have amended the following sentence, explaining the reasoning for this behavior. It now reads:

*[...] The reason for this might either be the focus of the unsupervised classification on re-frozen leads or the low resolution of SMOS, which does not allow to resolve isolated parts of thicker drift ice.*

We are not using the SMOS sea ice thickness uncertainty information here, because the uncertainty in the product files is an estimate of the retrieval uncertainty and almost directly related to sea ice thickness [Reference below]. Rather than trying to find a relationship between SMOS uncertainty and CryoSat-2 classification error, we will attempt to use the CryoSat-2 classification to inform the SMOS retrieval in future work.

SMOS L3 Sea Ice Thickness ATBD, ESL for Geophysical (L3 & 4) data over land ice and sea ice, https://earth.esa.int/eogateway/documents/20142/0/SMOS-L3-Sea-Ice-Thickness-ATBD

---

## Author Comment (AC2)

We thank the Reviewer for the careful and constructive comments. The suggestions and corrections have greatly improved the quality of this manuscript.

Summary

The authors extend an existing unsupervised classification approach for detecting open water targets from radar altimeter waveforms, to include detection of thin-ice. The authors present the classification approach conceptually, apply it to Cryosat-2 L1B Ice Baseline D data waveforms then compare their results to existing thin ice detection techniques from passive microwave radiometry (SMOS sensor) and infrared data (MODIS sensor), as well as Sentinel-1 SAR imagery. A comparison is also made to another Cryosat-2 sea ice product, the AWI Cryosat-2 sea ice product v2.4. The Laptev Sea is the region of focus due to the occurrence of winter period thin ice areas from flaw leads and coastal polynyas. The classification is applied to Cryosat-2 data taken January through March over the 2011 to 2020 period. Overall, the method is promising for detection of thin-ice areas, as a potential improvement over the ESA Climate Change Initiative (CCI) algorithm described in Paul et al. (2018) which does not include this feature. With a better evaluation in comparison to the prepared MODIS thin-ice thickness data (see major comment below), the utility and limitations of the proposed algorithm will be better understood. The work is clearly justified, and as the authors state, knowledge of thin ice areas is important for sea ice mass balance work, and there are limited datasets for remote study of their properties and coupled processes. It should be a good contribution to TC and of interest to the readership after comments here, and in the other reviews, are adequately addressed.

Major comments:

1. The authors compiled a large number of overlaps between their classification and MODIS TIT information. Despite this, the use of MODIS TIT to understand classifier performance is limited to a brief statement about overall class assignments on lines 267-269. The MODIS TIT data are further used for assessing relationships between ice thickness categories and waveform properties, which adds value to the interpretation of classifier performance. However the paper would be much improved by using the MODIS TIT for more detailed comparisons to the UWC classification outputs, in order to better understand limitations and inform further classifier optimization. The comparison to the CCI data is much more informative.

   Unfortunately, we cannot make the same comparisons between MODIS and UWC as we did between CCI and UWC, because the data come from different sensors and have different resolution. We are comparing two sensors with different resolution with CryoSat-2 and MODIS. A CryoSat-2 observation classified as sea ice in a MODIS pixel with thin ice may be a correct classification if small amounts of sea ice were present there or if the situation changed between the MODIS and CryoSat-2 images. For this reason a more detailed quantitative comparison is therefore not meaningful in our opinion.

2. The visual comparisons between the classification and coincident sensor data are helpful, and the authors put together a good summary of relationships between observed conditions in each comparison. The authors should provide a clear rationale for the choices made, in order to add confidence that these are un-biased assessments. It is apparent that the Cryosat-2, MODIS, and

Sentinel-1 comparison on 01 March 2018 is chosen due to the short time gap between all three acquisitions. The other comparisons are not as well justified, and it is likely there are several overlaps to choose from.

As the reviewer pointed out and it is also described in the manuscript, the triple comparison (CS-2, MODIS, S1) was chosen for the small time gap but as well for it being the only of its kind found in the analyzed time period. All other shown comparisons were chosen for the especially small time gaps and/or clear-sky conditions between acquisitions. However, all analyzed pairs of MODIS and CS-2 data were acquired within a 30 minute time gap as a minimum requirement for the comparison due to otherwise fast changing sea-ice conditions (see L61). We added the respective acquisition time differences to the respective figure captions.

Minor Comments (by line number):

L4: clarify that it is a Cryosat-2 based classification here

We agree.

L6: clarify what linear dependency is found (e.g. "between…x and y…") or consider re-wording

We clarified the sentence as follows:

*"Here, strong linear dependencies are found between binned thin-ice thicknesses up to 25cm thickness from MODIS and the CryoSat-2 waveform shape parameters that show the possibility to either develop simple correction terms for altimeter ranges over thin ice or to directly adjust current retracker algorithms specifically to very thin sea ice."*

L22: delete "in their retrieval capabilities as well as" and replace with "and"

We agree.

L26: comma after "sensors"

We agree.

L29: delete "are also prone to"

We agree.

L36: commas after "density" and "cover"

We agree.

L38: delete "But even when … classified as sea ice"

We thank the reviewer for his suggestion. However, we rephrased the corresponding sentence as follows:

*"For correct sea-ice classifications, the small freeboard values of thin ice (here defined as sea ice with a thickness up to 25 cm) are often lower than the precision of even the later synthetic aperture radar (SAR) altimeter sensors."*

L39: change "the later" to "recent" and "over" to "of" (i.e. freeboard *of* ice)

We agree.

L40: "from" Ku-band radar altimeters

We agree.

L44: delete "since the range …depth."

We agree.

L48: delete "on"

We rephrased the sentence and deleted "on".

L49: delete "short comings and"

We agree.

L51: Make "However, at a lower spatial resolution …" a new sentence

We agree.

L58: use "spatial and temporal resolutions" or "spatio-temporal resolution"

We agree and switch to spatio-temporal resolutions.

L65-68: "This study is structured into the following sections: Section 2 describes the data sets; Section 3 provides details on the unsupervised clustering for CryoSat-2 and the MODIS thin-ice thickness retrieval. Section 4 summarizes and discusses the results and implications on CryoSat-2 surface-type classification, and Section 5 concludes with an outlook."

Thank you for the re-structuring of the sentence.

L70: "The following sub-sections highlight the data sets used …"

We agree.

L74-75: delete "aiming at monitoring …was placed"

We think that this sentence is a good introduction for the section "CryoSat-2 Level-1B Baseline-D data". This sentence provides the reader with some basic information about the main science goals of CryoSat-2 and serves as an introduction to the text that follows. Therefore, we decided to leave it in the manuscript.

L76: delete "Moreover,"

We agree.

L79: delete "mainly characterized … ice cover,"

We agree.

L80: change "showing" to "with"

We agree.

L85: "This dataset comprises, …"

We agree.

L86: comma after data, and delete "also", and add "on" after information
We agree.
L91: delete "As basis" and just use "MODIS" rather than writing it out

We agree.

L92: delete "MODIS sensors on board the polar orbiting"

We agree.

L93-94: The MODIS data access information can be moved to the data availability section

We agree and moved it to the data availability section

L96: delete "In a first step"

We agree.

L102: Add information about the temporal component, e.g. hourly data nearest to acquisition, daily average, etc.

We thank the reviewer for pointing this out to us and rephrased the sentence as follows:

*"These fields comprise the 2 m air temperature, the 10 m wind-speed components, the mean sea-level pressure, and the 2 m dew-point temperature in hourly resolution."*

L110: "and less rough surface under calm, low-wind, conditions." The possibility of wind-roughened polynya and lead should be mentioned since this would also contribute strong backscatter and bright pixel values.

We add a sentence. Thank you for the hint.

L112: "e.g. on nilas ice, as they…"

We agree.

L115-116: move the introduction to Sentinel-1 to beginning of the section

We agree.

L118-120: Since the Sentinel-1 data are used only for visual assessment, the processing steps are likely straightforward and could be described briefly here.

We add a sentence. However, it must be really mentioned that only standard methods, provided by the SNAP Toolbox, are applied.

L123-125: delete the first sentence

The sentence is intended to softly introduce the reader to the subject matter of Section 3. We would therefore like to keep the sentence.

L127: change "keeping" to "enabling"

We agree.

L129: delete "used"

We agree

L130: clarify what is meant by a Cryosat-2 observation

We add "altimeter".

L131: sentence "The number of useable …" is not necessary because of previous descriptions

We agree and removed the sentence.

L139: delete "has"

We agree

L143: put a comma after "clustering" and delete "next classification steps consist of an"

We rephrased the sentence.

L148: delete "so called" and "followed"

We partly agree with the reviewer and deleted "so-called". However, we keep "followed" since it introduces the subsequent list.

L163" here, and elsewhere, just use the abbreviation after it has been defined (i.e. MP)

We changed that. However, we kept the full-name in the captions of figures and tables.

L167: nilas

We agree

L168: delete "of"

We agree

L170: also compare to wind-roughened water

We have found that thin-ice reflections are very close to lead-like reflections. However, with less MP and a slightly wider Wwidth. This is especially true when leads have recently frozen over and a thin layer of ice has formed. Nevertheless, we point out in the Outlook that further research is needed to find out how the thin ice in combination with different wind speeds affects the waveforms. However, this is not part of the current study.

L176: as mentioned above for MP, just use "IST" since it was defined earlier

We changed all occurrences of 'ice-surface temperature' as well as 'sea-ice-surface temperature' to 'IST' after it was first defined as an abbreviation.

L191: Change text to "The very high spatial and temporal resolution altimetry …."

We agree. In reference to the previous comment, we agreed on "The very high spatio-temporal resolution altimetry …"

Figure 2 caption: use italics for words in quotes (and delete quotation marks)

We agree.

L199: use "subsets" in place of "zoomed-in snippets"

We agree.

L200: pluralize to "results"

We agree.

L207: delete "and, therefore, no truth is available"

We agree.

L207-209: provide some indication of what is meant by lead or thin-ice surface being very small (how small)

Added ", i.e. less than 1% of the illuminated surface (Drinkwater et al., 1991)"

L215: comma after "Northwards"

We agree.

L216: delete "in ice thickness"

We agree.

L222: delete "in a quantitative analysis"

We agree.

L223: change "acquired at" to "from"

We agree.

L224: delete "from each other"

We agree.

L229: recommend to change "supposedly" to "likely"

We agree.

L233: How is thin ice labelled correctly as lead and not thin ice?

This area is very close to 0 cm in the MODIS TIT plot. This region is a mixture of small thin ice patches and open water spots. The spatial resolution of MODIS, however, makes it impossible to distinguish between these surface properties on a finer scale. The CryoSat-2 altimeter observations, however, are superimposed on the much stronger backscattering lead reflections in this region, thus the waveforms are considered as lead returns.

L239: change "blended with" to "compared to"

We agree.

L270: "TIT"

We agree.

L272: "TIT"

We agree.

L273: delete "spatiotemporally"

We agree.

L276: "deviations"

We agree.

L286: delete "However"

We agree.

L287: change "this" to "a"

We agree.

L293: change "Contrariwise" to "In contrast" and use "LEP" only since it is already defined

We agree.

L304: citation needed

We thank the reviewer for this comment. We added a reference instead of the placeholder.

L309: use LEW

We agree.

L331: delete comma after "both"

We agree.

L334: "(WMO, 2014)"

We agree.

L346: delete "in general"

We agree.

L350: delete "in general" and comma after "both"

We agree.

---

## Referee Report (RR1)

**Monitoring Arctic thin ice: A comparison between Cryosat-2 SAR altimetry data and MODIS thermal-infrared imagery, R1**

Felix L. Müller, Stephan Paul, Stefan Hendricks, and Denise Dettmering

**21 Nov 2022**

The authors have presented detailed, proper answers to most of my comments to the first version of the paper, and made corresponding changes and additions to the paper. I think that the paper has improved considerably. Below I have some minor comments for your consideration for further possible paper improvements.

"It is true that the SMOS dataset is not mentioned in the datasets. We would like to focus intentionally on Cryosat-2, MODIS and Sentinel-1, since these data were mainly used. In the case of SMOS, a readymade data product was used. The dataset was taken unchanged as a basis for comparison and is therefore well documented and referenced, whereas in the case of Cryosat-2, MODIS and Sentinel-1 observations had to go through various pre-processing steps."

I have to strongly disagree here, you should describe the SMOS dataset under Section 2.

line 38: "Numerous studies exist on the automatic detection of leads. These use altimeter data (e.g., Lee et al., 2018; Dettmering et al.; Wernecke and Kaleschke, 2015) or SAR images (e.g., Park et al., 2020; Boulze et al., 2020)."

Please include also Murashkin et al., 2018 here.

"This algorithm was developed at UB and AWI, so maybe the authors have access to it?"

You could discuss in the paper why did you not use automatic methods to detect leads/thin ice in SAR images.

Reference details missing:

Dettmering, D., Wynne, A., Müller, F. L., Passaro, M., and Seitz, F.: Lead Detection in Polar Oceans—A Comparison of Different Classification Methods for Cryosat-2 SAR Data, Remote Sensing.

"However, none of the existing studies has yet attempted to detect thin-ice using satellite altimetry data."

I guess I have to trust you here; I don't have time to do a survey on altimeter literature, and I have not worked much with altimeter data.

"Only manual visual screening was conducted as the MODIS cloud mask tends to eliminate especially very thin ice areas"

"Is snow on thin ice neglected in the TIT calculation?"

"Yes. This is one assumption in general for the retrieval of thin-ice thickness from MODIS in our and also other approaches and also outlined in, e.g., Paul et al (2015)."

Please give this information (snow and cloud screening) in Section 3.2.

"From ERA5 2m air temperature data, the average temperatures for the study period and region is always well below freezing point as one would expect in winter (about 253K depending on the exact location; a). However, there are rare occasions of above freezing-point temperatures (about 1.6%; b) especially over land, but also near the coast over sea-ice/fast-ice areas. While certainly the surface conditions change under these conditions and impact the received returns for CryoSat-2, we consider the overall impact on the UWC negligible."

Please give this discussion on weather conditions in the paper.

"[...] the basic ESA Climate Change Initiative's (CCI) CryoSat-2 surface-type classification with classes 'sea ice', 'lead', and 'unknown' [...]"

In the track changes doc 'sea ice', 'lead', and 'unknown' are missing.

"[...] there is currently only a single operational Arctic-wide thin-ice data product available [...]"

This is not exactly the case: there are two SMOS SIT products, one by UHAM (now AWI?) and one by UB, using same SMOS input data, but different algorithms. You should also give references to UB product.

"We follow the reviewer's suggestion and provided numerical (Pearson) correlations. Please find a table in the Appendix."

Yes, good to have these correlations, but you should also include some discussion on them to the text.

"We have added the sea ice concentration source products to the manuscript."

Also references or web-links should be given.

1. 24: "Leads and polynyas are openings of varying size and shape between drift..."

Between 'drift ice'? Leads and polynyas can also occur within landfast ice.

1. 69: "resolution to enable future monitoring of thin ice with a high spatial-temporal resolution"

You could define what mean by 'high spatial-temporal resolution', in context of which applications.

Start of Section 2.3 needs editing – two first sentences on Sentinel-1.

Finally,

"It would be interesting to also see how CS2 thin ice classification works as a function of MODIS TIT, a similar investigation is now conducted for CS2 waveform features. "

"This comment leads in the direction of the table shown in the comment above."

This table is interesting and you could consider to include it to the paper with some discussion. I understand that coarse MODIS resolution causes some errors here (explain this in the paper).

---

## Author Response (AR2)

**Reviewer 1**

**Monitoring Arctic thin ice: A comparison between Cryosat-2 SAR altimetry data and MODIS thermal-infrared imagery, R1**

Felix L. Müller, Stephan Paul, Stefan Hendricks, and Denise Dettmering

**21 Nov 2022**

The authors have presented detailed, proper answers to most of my comments to the first version of the paper, and made corresponding changes and additions to the paper. I think that the paper has improved considerably. Below I have some minor comments for your consideration for further possible paper improvements.

We thank the reviewer for the constructive support. All remaining technical comments were accepted and added to the manuscript. Individual responses to comments can be found in the following section.

"It is true that the SMOS dataset is not mentioned in the datasets. We would like to focus intentionally on Cryosat-2, MODIS and Sentinel-1, since these data were mainly used. In the case of SMOS, a readymade data product was used. The dataset was taken unchanged as a basis for comparison and is therefore well documented and referenced, whereas in the case of Cryosat-2, MODIS and Sentinel-1 observations had to go through various pre-processing steps."

I have to strongly disagree here, you should describe the SMOS dataset under Section 2.

With respect to the Editor comment and our initial focus on a MODIS/CS2 comparison, we decided to remove the SMOS comparison from the main manuscript body (as well as Figure 7) and therefore also eliminate the need for a more detailed description in Section 2 of the data. However, we added an additional Figure from our letter to the Editor into the manuscript's Appendix as an exemplary basis for our decision to not further investigate SMOS as a potential candidate for this type of comparisons. All occurrences in the manuscript were adjusted accordingly to reflect this decision.

line 38: "Numerous studies exist on the automatic detection of leads. These use altimeter data (e.g., Lee et al., 2018; Dettmering et al.; Wernecke and Kaleschke, 2015) or SAR images (e.g., Park et al., 2020; Boulze et al., 2020)."
Please include also Murashkin et al., 2018 here.

Done

"This algorithm was developed at UB and AWI, so maybe the authors have access to it?"

You could discuss in the paper why did you not use automatic methods to detect leads/thin ice in SAR images.

We added Murashkin et al. as a reference for SAR lead detection in the Introduction and an explanation for not including automatic SAR image processing in Section 2.3

*"An automatic thin-ice detection from SAR images is not applied in this study because this falls outside the main focus of the work."*

Reference details missing:

Dettmering, D., Wynne, A., Müller, F. L., Passaro, M., and Seitz, F.: Lead Detection in Polar Oceans—A Comparison of Different Classification Methods for Cryosat-2 SAR Data, Remote Sensing.

Done

"However, none of the existing studies has yet attempted to detect thin-ice using satellite altimetry data."

I guess I have to trust you here; I don't have time to do a survey on altimeter literature, and I have not worked much with altimeter data.

"Only manual visual screening was conducted as the MODIS cloud mask tends to eliminate especially very thin ice areas"

"Is snow on thin ice neglected in the TIT calculation?"

"Yes. This is one assumption in general for the retrieval of thin-ice thickness from MODIS in our and also other approaches and also outlined in, e.g., Paul et al (2015)." Please give this information (snow and cloud screening) in Section 3.2.

Done

"From ERA5 2m air temperature data, the average temperatures for the study period and region is always well below freezing point as one would expect in winter (about 253K depending on the exact location; a). However, there are rare occasions of above freezing-point temperatures (about 1.6%; b) especially over land, but also near the coast over sea-ice/fast-ice areas. While certainly the surface conditions change under these conditions and impact the received returns for CryoSat-2, we consider the overall impact on the UWC negligible."

Please give this discussion on weather conditions in the paper.

Done. Please see Section 3.

"[...] the basic ESA Climate Change Initiative's (CCI) CryoSat-2 surface-type classification with classes 'sea ice', 'lead', and 'unknown' [...]"

In the track changes doc 'sea ice', 'lead', and 'unknown' are missing.

This was a technical LaTex problem. We thank for that hint and can confirm that everything is included in the non-track change document.

"[...] there is currently only a single operational Arctic-wide thin-ice data product available [...]"
This is not exactly the case: there are two SMOS SIT products, one by UHAM (now AWI?) and one by UB, using same SMOS input data, but different algorithms. You should also give references to UB product.

We refer here to the official operational ESA SMOS product. However, we added a reference to the UB product as well.

"We follow the reviewer's suggestion and provided numerical (Pearson) correlations. Please find a table in the Appendix."

Yes, good to have these correlations, but you should also include some discussion on them to the text.

We have included correlations in the text to support the discussion.

"We have added the sea ice concentration source products to the manuscript." Also

references or web-links should be given.

We added a reference.

l. 24: "Leads and polynyas are openings of varying size and shape between drift…" Between

'drift ice'? Leads and polynyas can also occur within landfast ice.

Per definition the ice would not be land-fast anymore with an active polynya/lead within it but at least partly mobile. We assume the reviewer refers to flaw leads/polynyas which occur at the fast-ice edge. However, there was an 'ice' missing after 'drift' and we corrected the sentence in the manuscript accordingly.

l. 69: "resolution to enable future monitoring of thin ice with a high spatial-temporal resolution"
You could define what mean by 'high spatial-temporal resolution', in context of which applications.

We rephrased this sentence to read: "[...] NASA's Moderate Resolution Imaging Spectroradiometer (MODIS) to allow for a better understanding of the received CryoSat-2 waveform returns over thin sea-ice areas and, subsequently, an improved surface-type classification."

Start of Section 2.3 needs editing – two first sentences on Sentinel-1.

We rephrased the two sentences and added information about the status of Sentinel-1B.

Finally,

"It would be interesting to also see how CS2 thin ice classification works as a function of MODIS TIT, a similar investigation is now conducted for CS2 waveform features. "
"This comment leads in the direction of the table shown in the comment above."

This table is interesting and you could consider to include it to the paper with some discussion. I understand that coarse MODIS resolution causes some errors here (explain this in the paper).

We prefer to leave this table in the rebuttal of Review 1, which is public and can be viewed at any time. There are various notes and discussions in the paper regarding the different resolutions of MODIS and CS-2.

**Reviewer 2**

**Suggestions for revision or reasons for rejection (will be published if the paper is accepted for final publication)**

The authors have made substantial improvements to the paper by thoroughly addressing several of the reviewer comments. In cases where the authors did not act on a reviewer comment, they provided a justification for their decision. The authors should be commended for their attention to detail in making the revised version of the manuscript a worthwhile contribution to The Cryosphere pending some additional minor comments and corrections.

We thank the reviewer for the constructive support. All remaining technical comments were accepted and added to the manuscript. Individual responses to comments can be found in the following section.

By line number.
L7: thin-ice thickness (singular not plural since it is a variable) : done
L9: change "or to directly adjust" to "or direct adjustments to" : done
L9: "… specifically to" should be "…. for" : done
L11: delete "mere" : done
L12: delete "basic" : done
L13" delete "between" : done
L20: delete "sea-ice" before leads and polynyas : done
L27: delete "the" : done
L38: change "allows …. measure" to "enables direct measurements of" : done
L52: This doesn't need to be a new paragraph. : done
L57: Since this is mentioning the current situation, it would be appropriate to update this to the combined SMAP-SMOS thickness product.

We refer here primarily to ESA's official operational product which is based on SMOS only but are also including now the University of Bremen's similar data product due to a strong request by Reviewer 1. The combined product, however, is not considered on par and also not used due to its very limited availability.

L64: delete "the capabilities of" : done
L79: should be "highlights" : done
L83-84: It would be better to be consistent with ESA stated primary Cryosat-2 objectives of measuring diminishing land and sea ice in polar regions, for understanding the role of climate in diminshing polar ice.

We thank for this comment and changed the sentence.

*CryoSat-2 was launched in April 2010 aiming to monitor the Earth's cryosphere and, in particular, to measure the decline of land and sea ice in the polar regions in order to understand the role of climate in the retreat of polar ice.*

L85: delete "the cryosphere": done
L97: change waveform returns to "waveforms" : done
L102: Clarify the wavelengths associated with these band numbers. : done
L111: Since Sentinel 1B is no longer operative, it would be appropriate to update this sentence to better reflect the situation. Or change it to past tense to reflect the situation relevant to the study.

*We added a sentence.*
*Unfortunately, Sentinel-1B became non-operational after an instrument failure in December 2021.*
L115: provides: done
L116: delete "in the ice covered ocean": done
L121: "a" shows up twice: done
L128: map projection (no "s"): done
L129: It is unclear what is meant by theoretical size. If this is an estimate it should be indicated as such.: We mean the potentially usable image dataset size. We changed that in the manuscript.
L142: Please clarify what is meant by observations here. It is relatively straightforward but someone outside of altimetry might find it difficult to follow whether an observation is a track or a waveform (or other). : done
L157: Delete "see". Also "Fig." or "Figure" be consistent: done
L191: It would be more appropriate to define this when first used as TIT and use TIT throughout.: : done
L202: Confusing sentence. Do you mean "… NCEP2 reanalysis by Kalnay et al. … "?: yes – this was a LaTeX typo and we changed it to show the reference in parenthesis
L204: TIT: done
L209: Since comparisons are also made to SMOS, it would be better to mention that here too. Currently it is on L217.
The SMOS comparison was removed from the manuscript to focus on the initial CS2/MODIS comparison and replaced by an additional Figure taken from our Letter to the Editor in the previous round of revisions.

L225: Where it is mention that reasons are two-fold, some consideration of the geometric i.e. positional accuracy of the different sensing systems should also be considered here. At least nominally, how well is everything co-located?

We are not quite sure if we understand the comment correctly or what the reviewer is aiming at. The orbit accuracy of both missions is in the lower centimeter range. We consider these smaller uncertainties to be negligible in comparison to other influences such as co-occurring sea ice drift or the size of the SAR pixels as well as footprint coverage of the altimeter.

L228: add a space after small: done
L243: Unclear what is meant by "fall up". Contain?: We changed to "contain".
L255: "… change in roughness …": done
L257: Change "But is" to "It" : done
L262: Delete "really" : done
L275: change "but also" to "and: we totally rephrased this part
L279: It would be helpful to have some context on expected growth rates of sea ice, or at least how long it takes for a lead to freeze over, with reason given that it will be dependent on e.g. air temp, water temp, etc. (minutes, hours, days, etc.)?

While the mentioned paragraph was removed in the context of eliminating the SMOS comparison in this study, a simple answer as mentioned also by the reviewer is difficult to provide. The growth rate depends on several parameters (air temp, ocean heat flux/water temp, salinity, etc.). However, as outlined in Notz and Worster (2008; Figure 7, included below), within about 24 hours the ice growths up to between 5-8 cm, which is slightly above the MODIS uncertainty provided for this study for a maximum time difference between CryoSat-2 and MODIS acquisitions.

[Figure]

*Notz, D., and Worster, M. G. (2008), In situ measurements of the evolution of young sea ice, J. Geophys. Res., 113, C03001, doi:10.1029/2007JC004333.*

Figure 8 caption: the average feature values are blue circles not "blue dotted line": done[1]
L382: Is it noise, or just a more variable signal (less peakiness)?:

Both signal characteristics are included. The noise increases with a simultaneous decrease in peakiness. We have added "higher variability" and "less peakiness".

L391: For the reason stated in the sentence beginning "Moreover …" the summary is better and stronger without this in it
This was removed in the process of eliminating the SMOS comparison from the current version of the manuscript.
* * *
[1] Please note: Now it is Figure 7